METHODS AND RESOURCES

# A light-inducible protein clustering system for in vivo analysis of α-synuclein aggregation in Parkinson disease

Morgan Bérard[1,2], Razan Sheta[1,2], Sarah Malvaut[3,4], Raquel Rodriguez-Aller[1,2,3], Maxime Teixeira[1,2], Walid Idi[1,2], Roxanne Turmel[1,2], Melanie Alpaugh[1,4], Marilyn Dubois[1,2], Manel Dahmene[1,2], Charleen Salesse[3,4], Jérôme Lamontagne-Proulx[1,5], Marie-Kim St-Pierre[1,2], Omid Tavassoly[6], Wen Luo[6,7], Esther Del Cid-Pellitero[6,7], Raza Qazi[8], Jae-Woong Jeong[8,9], Thomas M. Durcan[6,7], Luc Vallières[1,2], Marie-Eve Tremblay[1,2,10], Denis Soulet[1,5], Martin Lévesque[3,4], Francesca Cicchetti[1,4], Edward A. Fon[6,7], Armen Saghatelyan[3,4], Abid Oueslati[1,2]*

1 CHU de Québec Research Center, Axe Neurosciences, Quebec City, Canada, 2 Department of Molecular Medicine, Faculty of Medicine, Université Laval, Quebec City, Canada, 3 CERVO Brain Research Centre, Quebec City, Canada, 4 Department of Psychiatry and Neurosciences, Faculty of Medicine, Université Laval, Quebec City, Canada, 5 Faculty of Pharmacy, Université Laval, Quebec City, Canada, 6 McGill Parkinson Program and Neurodegenerative Diseases Group, Montreal Neurological Institute, McGill University, Montreal, Canada, 7 The Neuro's Early Drug Discovery Unit (EDDU), Montreal Neurological Institute, McGill University, Montreal, Canada, 8 Department of Electrical, Computer, and Energy Engineering, University of Colorado, Boulder, Colorado, United States of America, 9 School of Electrical Engineering Korea Advanced Institute of Science and Technology, Daejeon, Republic of Korea, 10 Division of Medical Sciences, University of Victoria, Victoria, Canada

☯ These authors contributed equally to this work.
* abid.oueslati.1@ulaval.ca

**Data Availability Statement:** All relevant data are within the paper and its Supporting Information files.

## Abstract

Neurodegenerative disorders refer to a group of diseases commonly associated with abnormal protein accumulation and aggregation in the central nervous system. However, the exact role of protein aggregation in the pathophysiology of these disorders remains unclear. This gap in knowledge is due to the lack of experimental models that allow for the spatiotemporal control of protein aggregation, and the investigation of early dynamic events associated with inclusion formation. Here, we report on the development of a light-inducible protein aggregation (LIPA) system that enables spatiotemporal control of α-synuclein (α-syn) aggregation into insoluble deposits called Lewy bodies (LBs), the pathological hallmark of Parkinson disease (PD) and other proteinopathies. We demonstrate that LIPA-α-syn inclusions mimic key biochemical, biophysical, and ultrastructural features of authentic LBs observed in PD-diseased brains. In vivo, LIPA-α-syn aggregates compromise nigrostriatal transmission, induce neurodegeneration and PD-like motor impairments. Collectively, our findings provide a new tool for the generation, visualization, and dissection of the role of α-syn aggregation in PD.

**Funding:** This work was supported by Parkinson Society Canada, Fondation du CHU de Québec and the Canadian Institutes of Health Research (CIHR) grants to AO. AO was supported by Junior1 and Junior 2 salary Awards from the Fonds de Recherche du Québec - Santé (FRQS) and la Société Parkinson du Québec. In vivo Ca2+ imaging experiments were supported by the Canadian Institutes of Health Research (CIHR) grant to AS. MB was supported by scholarships from the Fondation du CHU de Québec, Faculty of Medicine of Université Laval (Bourse de recrutement du doctorat Pierre J. Durant), and FRQS. FC is a recipient of a Researcher Chair from the Fonds de Recherche du Québec en Santé (FRQS) and received funding from the Canadian Institutes of Health Research (CIHR). MA is supported by post-doctoral fellowships from both CIHR and FRQS. E.A.F. is supported by a Foundation grant from CIHR (FDN-154301) and a Canada Research Chair (Tier 1) in Parkinson's disease. MKSP was supported by a Frederick Banting and Charles Best Canada Graduate Scholarship-Doctoral Award and a Doctoral's training scholarship from FRQS. MET is a Tier 2 Canada Research Chair in Neurobiology of Aging and Cognition. TMD is supported with funds from the McGill Healthy Brains for Healthy lives and a project grant from CIHR (PJT–169095). The funders had no role in study design, data collection and analysis, decision to publish, or preparation of the manuscript.

**Competing interests:** The authors have declared that no competing interests exist.

**Abbreviations:** AAV, adeno-associated virus; α-syn, α-synuclein; APP, amyloid precursor protein; β-syn, β-synuclein; BSA, bovine serum albumin; CLEM, correlative light-electron microscopy; CNS, central nervous system; DAB, 3,3'-diaminobenzidine tetrahydrochloride; DAPI, 4,6-diamidino-2-phenylindole; DLB, dementia with LB; DMEM, Dulbecco's modified Eagle's medium; FBS, fetal bovine serum; hiPSC, human-induced pluripotent stem cell; LB, Lewy bodies; LIPA, light-inducible protein aggregation; LUV, large unilamellar-like vesicle; mα-syn, mouse α-syn; NAC, non-amyloid-β component; PBS, phosphate-buffered saline; PD, Parkinson disease; PDD, PD with dementia; PFA, paraformaldehyde; Pff, preformed fibril; PMCA, protein misfolding cyclic amplification; PMSF, phenylmethylsulfonyl fluoride; ROI, region of interest; RT-QuIC, real-time quaking-induced conversion; SNc, substantia nigra pars compacta; STED, stimulated emission depletion; SUV, small unilamellar-like vesicle; TEM,

# Introduction

Protein aggregation is a process by which misfolded proteins adopt an organized and structurally well-defined fibrillar conformation leading to the formation of proteinaceous amyloid deposits [1]. This process is established via a nucleation-dependent polymerization mechanism, also referred to as the seeding effect, whereby small oligomers provide a template for the assembly of soluble monomers into highly ordered protein aggregates defined by their insolubility and β-sheet structure [2]. One example is the accumulation of intraneuronal protein inclusions called Lewy bodies (LBs) in the brains of patients suffering from α-synucleinopathies, a group of neurological disorders that encompasses Parkinson disease (PD), PD with dementia (PDD), and dementia with LB (DLB) [3]. These inclusions are predominantly composed of aggregated α-synuclein (α-syn), a small protein ubiquitously and abundantly expressed in the brain [3, 4].

Since the initial description of LBs was published more than a century ago [5,6], a causal link between LBs formation and neurotoxicity has been suggested [3,6]. However, the exact roles of these inclusions in the pathogenesis and progression of PD and related disorders remain elusive [7]. This gap in knowledge is mainly due to a major limitation associated with the current α-syn overexpression-based experimental models of PD, which lack the temporal control and visualization of early aggregation steps. This limitation precludes the characterization of the primary dynamic events associated with α-syn inclusion formation and the investigation into the subsequent cellular repercussions [8–10]. Therefore, development of an experimental model of α-syn aggregation that allows for the precise spatial and temporal regulation of LB-like inclusion formation and for monitoring the early stages of this cellular process is urgently needed.

Recently, optogenetic control of protein behavior, known as optobiology, has emerged as a promising approach to studying molecular mechanisms that underlie biological processes with a high spatiotemporal resolution in living cells [11]. This approach is based on the use of engineered photoreceptors that change conformation in response to light and thereby influence the functional output of proteins that have fused together [11]. In the current study, we used optobiology to develop a new light-induced model of α-syn aggregation in cell culture and in the mouse brain. This model allows for real-time monitoring of α-syn aggregation and authentic LB formation with a high spatiotemporal resolution. Using this system, we demonstrate that light-induced α-syn aggregation affects neuronal homeostasis, leading to neurodegeneration and induction of PD-like motor impairments. Furthermore, by performing combined light-induced α-syn aggregation in the substantia nigra pars compacta (SNc) and mini-endoscopic calcium (Ca$^{2+}$) imaging in the striatum, we have identified dysfunction of nigrostriatal dopaminergic transmission at the early stages of α-syn aggregation. This model will help advance our understanding of how protein aggregation can affect neuronal homeostasis and will serve as an innovative tool to develop new anti-aggregation strategies as therapeutic targets for the treatment of PD and related α-synucleinopathies.

# Results

## The LIPA system allows for long-term induction of α-syn aggregation in living cells

Our approach is based on the use of the mutant form of *Arabidopsis thaliana*'s cryptochrome protein 2 (CRY2olig), a construct that undergoes rapid, reversible, and robust protein clustering when stimulated with blue light [12,13]. We hypothesized that when fused to α-syn, light-inducible CRY2olig clustering would trigger the aggregation of α-syn and thereby prompt the

transmission electron microscopy; ThS, thioflavin S; ThT, thioflavin T; WT, wild-type.

formation of LB-like inclusions in living cells. We referred to this system as light-inducible protein aggregation (LIPA) system. We subsequently engineered a construct in which human α-syn was fused to CRY2olig and mCherry (LIPA-α-syn) (Fig 1A). We also generated control constructs CRY2olig-mCherry (LIPA-Empty) and CRY2olig-mCherry fused to a nonaggregatable form of α-syn, missing the non-amyloid-β component (NAC) region (LIPA-α-syn$^{\Delta NAC}$) [14] (Fig 1A). To obtain equal protein expression levels for the different LIPA constructs, the amount of DNA used for the transfection of each plasmid was adjusted (0.5 μg of LIPA-Empty, 0.6 μg LIPA-α-syn, and 1 μg LIPA-α-syn$^{\Delta NAC}$ per well in a 6-well plate) and the mCherry expression levels evaluated by western blot (S1A Fig, S1 Data, S1 Raw images).

To investigate the capacity of the LIPA system to induce protein aggregation under prolonged light stimulation, we assessed the proportion of HEK-293T cells exhibiting mCherry-positive inclusions at different time points in a 24-hour period of continuous light stimulation. We stimulated the cells using blue light at an intensity of 0.8 mW/mm$^2$, which is below the phototoxicity threshold [12] (S1B Fig, S1 Data). Upon blue light stimulation, the proportion of cells with LIPA-Empty inclusions increased with time and reached a 50% plateau after 8 hours of stimulation (Fig 1B and 1C, S1 Data). Interestingly, cells overexpressing LIPA-α-syn exhibited faster and more abundant aggregate formation, showing a peak at 4 hours of stimulation and rapidly reaching an 80% plateau, suggesting that the presence of α-syn may enhance CRY2olig clustering (Fig 1B and 1C, S1 Data). Notably, no inclusions were observed at similar time points when cells were maintained in the dark, indicating that the LIPA system did not induce spontaneous aggregation in the absence of light stimulation (S1C Fig). Moreover, no aggregation was observed in cells overexpressing α-syn-mCherry, in the presence or absence of light stimulation, thus confirming that the mCherry tag did not induce aggregation per se (S1D Fig). Intriguingly, cells overexpressing LIPA-α-syn$^{\Delta NAC}$ did not exhibit visible inclusions, despite continuous light stimulation (Fig 1B and 1C, S1 Data). Given the fact that CRY2olig clustering efficacy is highly influenced by the oligomeric properties of the fused protein, affecting its photoactivated quaternary structure [15], the absence of LIPA-α-syn$^{\Delta NAC}$ aggregates may be associated with an inhibitory impact of mutant α-syn on CRY2olig clustering properties. We tested and confirmed this hypothesis by overexpressing LIPA-α-syn$^{\Delta NAC}$ in the presence of cryptochrome-interacting basic-helix–loop–helix 1 (CIB1), a protein known to interact with CRY2olig, to change its conformation and to facilitate its clustering upon blue light stimulation [16]. Upon exposure of these cells to blue light for 2 hours, we observed the formation of LIPA-α-syn$^{\Delta NAC}$ inclusions, which colocalized with CIB1, confirming that the overexpressed CIB1 interacted with CRY2olig and rescued the clustering capacity of CRY2olig (S2 Fig, S1 Data).

Then, we biochemically validated the formation of LIPA aggregates using a filter retardation assay, a technique that allows size- and solubility-based retention of high–molecular weight aggregates on the surface of a nonbinding acetate membrane. The results revealed the time-dependent accumulation of LIPA-α-syn and LIPA-Empty aggregates, while no aggregate retention was detected in the LIPA-α-syn$^{\Delta NAC}$ cell lysates (Fig 1D and 1E, S1 Data, S1 Raw images). Loading controls confirmed that a comparable protein amount was loaded under different experimental conditions (Fig 1F, S1 Data, S1 Raw images). Collectively, our data indicate that the LIPA system allows for long-term induction of α-syn aggregates exhibiting LB-like detergent insolubility.

## The LIPA system allows for real-time monitoring of α-syn aggregation with high spatiotemporal resolution

We evaluated the capacity of the LIPA system to allow monitoring of protein aggregation in HEK-293T cells using automated real-time live confocal imaging. Before light stimulation,

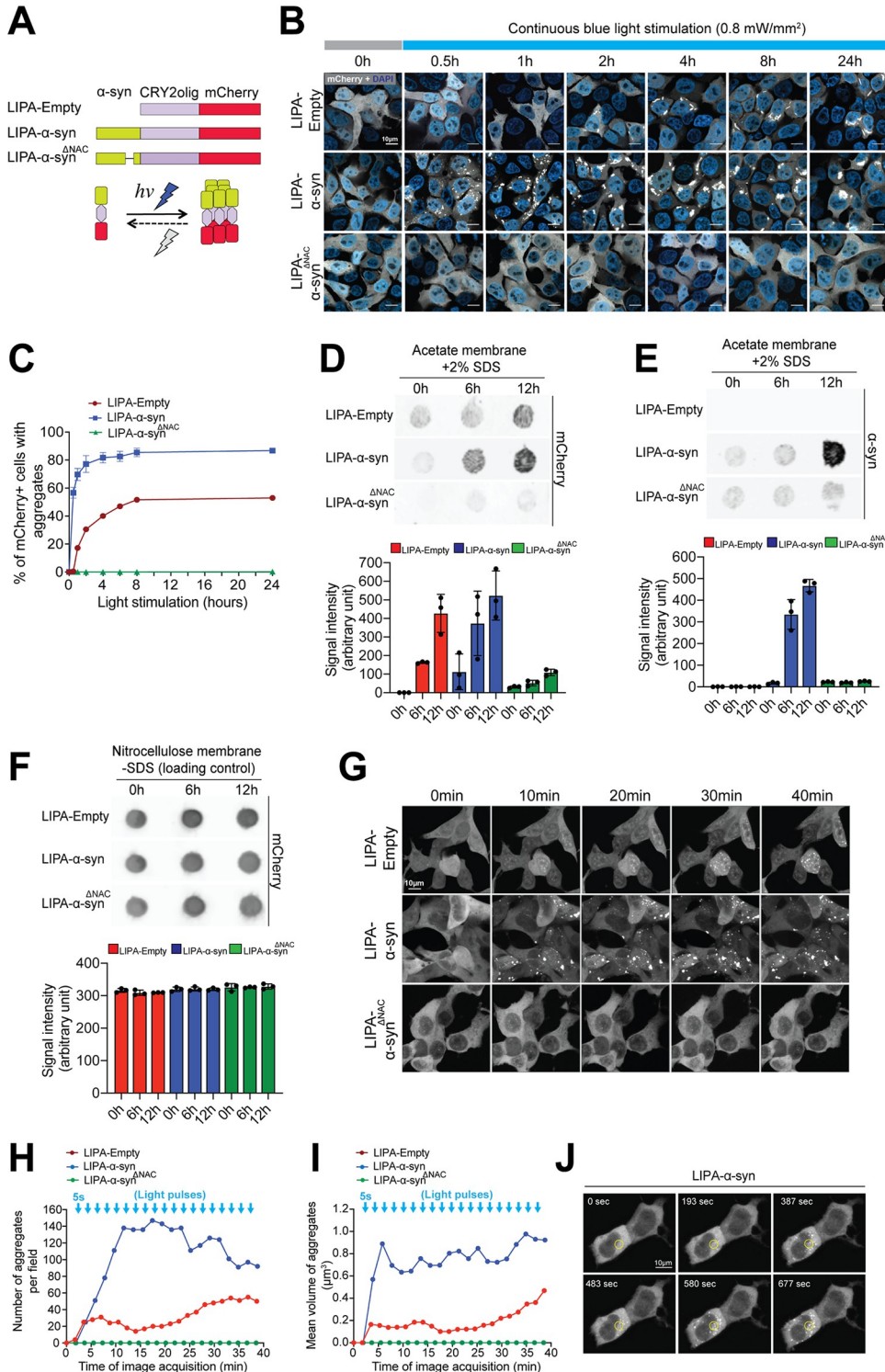

**Fig 1. The LIPA system allows for long-term induction and real-time monitoring of α-syn aggregation.** (**A**) Schematic representation of the LIPA constructs. Under blue light stimulation, the LIPA system induces robust and reversible protein clustering. (**B**) Representative confocal images of HEK-293T cells overexpressing the LIPA constructs exposed to continuous blue light stimulation (0.8 mW/mm²) for different time points up to 24 hours (scale bar = 10 μm). (**C**) Estimation of the percentage of mCherry-positive HEK-293T cells exhibiting LIPA aggregates at different time points during continuous blue light stimulation (220–250 cells/condition/time point; $n = 3$). The data are presented as the means ± SEM. (**D** and **E**) Filter retardation assay showing the time course of LIPA-Empty and

LIPA-α-syn aggregate formation as examined using mCherry and α-syn antibodies, respectively. The histograms present the aggregation signal intensity expressed in arbitrary units ($n$ = 3). The data are presented as the means ± SEM. (**F**) Prior to the addition of SDS, the samples were subjected to filtration through a nitrocellulose membrane, allowing the retention of monomeric and aggregate proteins and the verification of the total protein levels under each experimental condition. Histograms representing the levels of mCherry expression in arbitrary units ($n$ = 3). The data are presented as the means ± SEM. (**G**) Time-lapse live imaging illustration of representative HEK-293T cells overexpressing the LIPA constructs (scale bar = 10 µm). (**H**) Estimation of the total number of mCherry-positive aggregates and (**I**) the mean aggregate volume (µm³) after repetitive 5-second 488-nm light pulses (15.5 µW/mm²). (**J**) Time-lapse live imaging illustration of representative single-cell LIPA induction using 5-second 488-nm light pulses (15.5 µW/mm²) in an ROI (represented by the yellow circle) in HEK-293T cells overexpressing LIPA-α-syn (scale bar = 10 µm). The underlying data for (**C**), (**D**), (**E**), (**F**), (**H**), and (**I**) can be found in S1 Data. α-syn, α-synuclein; LIPA, light-inducible protein aggregation; ROI, region of interest.

LIPA constructs exhibited diffuse cytosolic expression (Fig 1G) and upon light stimulation, cells overexpressing LIPA-α-syn underwent rapid and robust formation of cytosolic inclusions, which appeared as large irregular mCherry-positive foci (Fig 1G, S1 Video). Importantly, due to CRY2olig protein clustering capacity, some cytosolic inclusions were also observed in cells overexpressing the LIPA-Empty construct (Fig 1G, S2 Video). However, the number and volume of aggregates were dramatically higher in cells overexpressing LIPA-α-syn (Fig 1H and 1I, S1 Data). This observation suggests that α-syn prompts light-induced clustering of CRY2olig. Notably, cells overexpressing LIPA-α-syn$^{\Delta NAC}$ did not depict inclusions, confirming that the presence of the nonaggregatable form of α-syn precludes CRY2olig cluster formation (Fig 1G-I, S3 Video, S1 Data).

We then validated the spatial resolution capacity of the LIPA system to prompt single-cell induction of LIPA-α-syn inclusions in HEK-293T cells. Live imaging revealed exclusive formation of LIPA-α-syn inclusions in the cytosol of the stimulated cell (the region of interest (ROI)) without inducing aggregate formation in neighboring cells (Fig 1J, S4 Video). Together, our results provide evidence that the LIPA system allows for the rapid induction and monitoring of α-syn aggregation in living cells with high spatiotemporal resolution.

## LIPA-α-syn inclusions recapitulate biochemical and ultrastructural features of authentic LBs in cell culture

We subsequently investigated whether light-induced LIPA-α-syn inclusions can reproduce key biochemical features of authentic LBs by using a battery of LB markers observed in α-synucleinopathy–diseased brains [17]. After 12 hours of continuous light stimulation, HEK-293T cells were subjected to immunocytochemistry and the results revealed that the LIPA-α-syn inclusions were positive for α-syn, phosphorylated α-syn (pS129), and ubiquitin, the principal neuropathological hallmarks of authentic LBs [18–20] (Fig 2A). Moreover, thioflavin S (ThS) staining revealed that LIPA-α-syn inclusions mainly contained β-sheet structures, a structural characteristic of authentic LBs [21] (Fig 2A). Finally, LIPA-α-syn inclusions were also immunopositive for p62 and HSP70, two chaperone proteins commonly observed within LBs in α-synucleinopathy–diseased brains [22] (Fig 2A). Interestingly, LB-like inclusions were also detected in human-induced pluripotent stem cells (hiPSCs)-derived neurons transiently overexpressing LIPA-α-syn and exposed to blue light for 6 hours (Fig 2B), demonstrating that the LIPA system allows for the formation of LB-like aggregates in different cell types, including postmitotic neurons. Of note, no LB-associated markers were observed in HEK-293T cells or in hiPSC-derived neurons overexpressing LIPA-α-syn in the absence of blue light stimulation (S3A and S3B Fig). Moreover, we did not observe any LB-related marker colocalizing with the LIPA-Empty inclusions or in the LIPA-α-syn$^{\Delta NAC}$ transfected cells, thus demonstrating that the LB markers are specifically associated with LIPA-α-syn aggregates (S4 and S5 Figs).

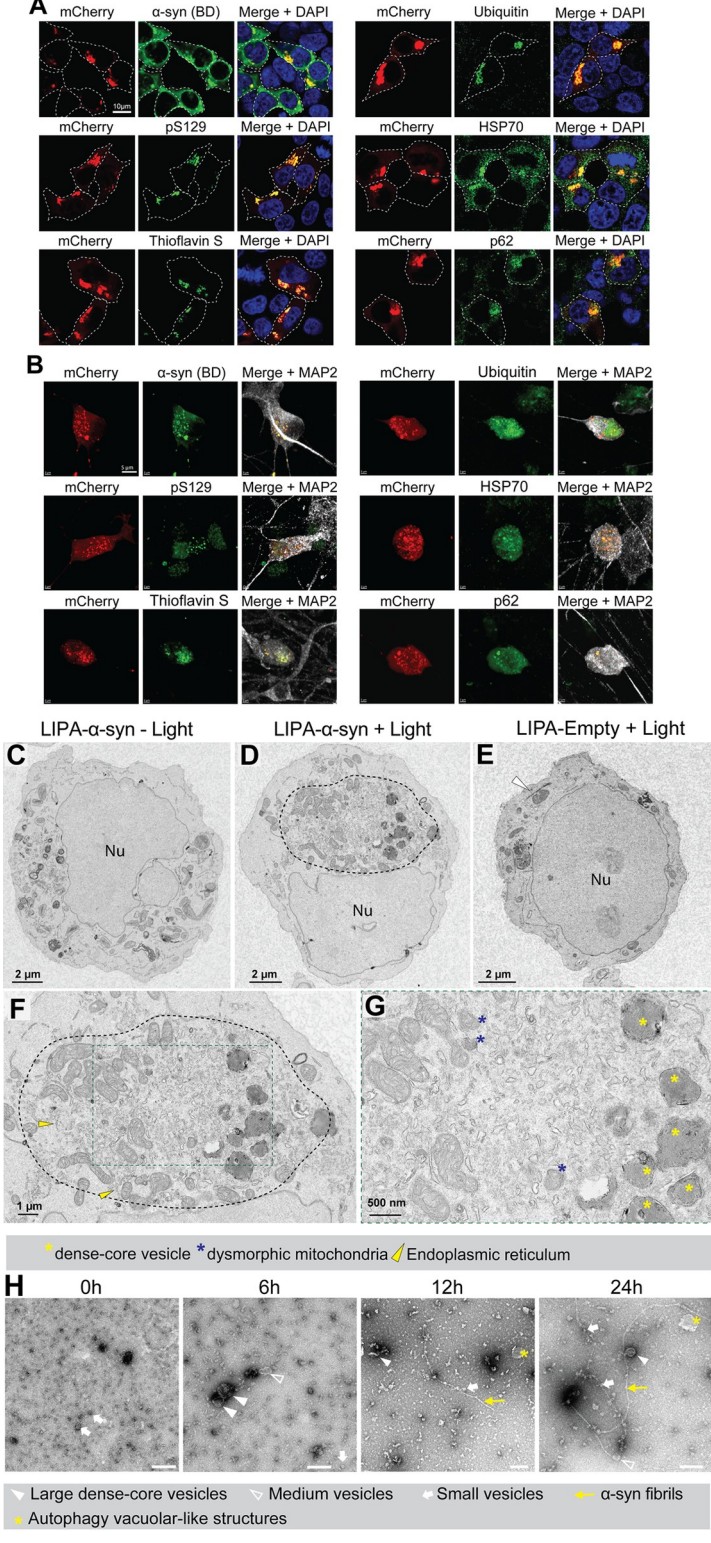

**Fig 2. LIPA-α-syn inclusions recapitulate authentic LB features in cell culture.** (**A**) Confocal images of HEK-293T cells overexpressing LIPA-α-syn exposed to blue light for 12 hours (0.8 mW/mm²) and stained with LB markers: α-syn (BDlab), phosphorylated α-syn at S129 (pS129), ThS, ubiquitin, HSP70, and p62 (*n* = 5) (scale bar = 10 μm). (**B**) Confocal images of hiPSC-derived human neurons overexpressing LIPA-α-syn exposed to blue light for 6 hours (0.4 mW/mm²) and stained with LB markers: α-syn (BDlab), phosphorylated α-syn at S129 (pS129), ThS, ubiquitin,

HSP70, p62, and MAP2 ($n$ = 3) (scale bar = 5 μm). (**C**) Electron micrograph of a cell overexpressing LIPA-α-syn and not exposed to the blue light (scale bar = 2 μm). (**D**) Electron micrograph of a cell overexpressing LIPA-α-syn and exposed to the blue light for 12 hours. The image illustrates the presence of circular structures (dashed line) corresponding to a LIPA-α-syn inclusion (scale bar = 2 μm). (**E**) Electron micrograph of a cell overexpressing LIPA-Empty and exposed to the blue light. The arrowhead points to the needle-like LIPA-Empty inclusion (scale bar = 2 μm). (**F**) At low magnification, LIPA-α-syn inclusion appear to be surrounded by mitochondria and endoplasmic reticulum (yellow arrowhead) and are mainly composed of abundant vesicular structures and dysmorphic organelles (scale bar = 1 μm). (**G**) At a high magnification, the images reveal the presence of dense core vesicles (yellow asterisk) and dysmorphic mitochondria (blue asterisk) (scale bar = 500 nm). (**H**) EM images of filamentous structure of α-syn from purified LIPA-α-syn aggregates from HEK-293T cells exposed to blue light for 0, 6, 12 and 24 hours (scale bar 0 hour = 500 nm; scale bar 6, 12, and 24 hours = 200 nm). α-syn, α-synuclein; hiPSC, human-induced pluripotent stem cell; LB, Lewy bodies; LIPA, light-inducible protein aggregation; Nu, nucleus; ThS, thioflavin S.

To determine the ultrastructure of the LIPA-α-syn aggregates, we performed nanoscale-resolution transmission electron microscopy (TEM). In HEK-293T cells overexpressing LIPA-α-syn, but not exposed to blue light, the organelles appeared intact and homogenously distributed in the cytosol (Fig 2C). However, when LIPA-α-syn cells were exposed to the blue light (12 hours), we detected a prominent formation of circular structures, with diameters ranging between 4 and 6 μm, located in the vicinity of the nucleus (Fig 2D). The size and the location of these structures suggest that they may correspond to LIPA-α-syn inclusions. Moreover, such circular structures were not observed in LIPA-Empty cells exposed to blue light (Fig 2E). However, we observed elongated structures (Fig 2E, arrowhead) resembling the needle-like structures of LIPA-Empty inclusions observed with confocal imaging (Fig 1B).

The ultrastructural analysis of high-resolution TEM images revealed that the circular structures observed in the LIPA-α-syn condition were surrounded by a ring of organelles, including mitochondria and the endoplasmic reticulum (Fig 2F and 2G). Moreover, the core of these structures was mainly composed of lipid vesicles and tubulovesicular structures of various sizes and shapes, as well as electron-dense vesicles with lysosome- and autophagosome-like structures, and distorted organelles, particularly dysmorphic mitochondria and fragmented endoplasmic reticulum. The composition of these structures closely resembled the description of authentic LBs recently reported using correlative light-electron microscopy (CLEM) [23].

Of note, under our experimental conditions, the TEM images did not show fibrillar structures within the LIPA-α-syn aggregates, probably due to the high density and crowdedness of these inclusions. However, when the aggregates were purified from HEK-239T cells exposed to blue light for 6, 12, and 24 hours, TEM analysis revealed the presence of such filamentous structures, which appeared to increase in length and thickness, in a light exposure time-dependent manner and interacting with lipids and vesicles, as previously reported [24,25] (Fig 2H).

To further identify the vesicles observed within LIPA-α-syn aggregates, we immunostained cells for endosomal markers and examined them using stimulated emission depletion (STED) microscopy. Images of these aggregates revealed the presence of early endosomes (EAA1), lysosomes (LAMP1 and LAMP2A), and exosomes (CD9) within LIPA-α-syn inclusions (S6 Fig). However, no colocalization of these vesicle markers were observed with LIPA-Empty inclusions nor in the LIPA-α-syn$^{\Delta NAC}$ condition (S6 Fig). Taken together, our TEM and STED data demonstrate prominent ultrastructural similarities between LIPA-induced aggregates and authentic LBs.

## The LIPA-induced α-syn inclusions mimic the seeding capacity of authentic LBs

A cardinal feature of pathological α-syn inclusions is their capacity to autoperpetuate through the recruitment of soluble monomeric counterparts, a process referred to as the protein-

seeding capacity [2,26]. To investigate this feature using the LIPA system, we first assessed the capacity of LIPA inclusions to be self-sustaining after transient optogenetic stimulation (12 hours) (S7A Fig). As previously reported, LIPA-Empty inclusions dissociated rapidly after light stimulation was terminated and virtually no inclusions were detected after the cells were kept in the dark for 48 hours [12] (Fig 3A). Remarkably, LIPA-α-syn inclusions were present in the majority of the cells 48 hours postillumination, suggesting that LIPA-α-syn inclusions can autoperpetuate and that α-syn might play a role in aggregate stability (Fig 3A, S1 Data). Treatment with small-molecule inhibitors of α-syn aggregation, namely, baicalein or myricetin [27,28], or the overexpression of β-synuclein (β-syn) or mouse α-syn (mα-syn), 2 proteins known to inhibit human α-syn aggregation [29–31], significantly reduced LIPA-α-syn inclusion stability, thus confirming the important role of α-syn in maintaining inclusion stability (Fig 3B and 3C, S1 Data). Notably, the same treatments did not affect the dissociation rate of LIPA-Empty inclusions (S7B and S7C Fig, S1 Data). Moreover, treatment with small molecules known not to have any effect on α-syn aggregation (naringenin or daidzein [27,28]) or the overexpression of GFP did not affect LIPA-α-syn inclusion stability (S7D Fig and S7E Fig, S1 Data). These observations demonstrate that LIPA-α-syn inclusion stability is indeed related to the presence of α-syn and to its aggregation capacity.

Next, we tested whether the stability of LIPA-α-syn inclusions is the result of the recruitment of soluble α-syn. To test this hypothesis, we assessed the capacity of LIPA-α-syn aggregates to recruit monomeric α-syn tagged with GFP in HEK-293T cells. Confocal imaging showed that before light stimulation, LIPA-α-syn and α-syn-GFP were diffused in the cytosol (Fig 3D). However, after 12 hours of blue light stimulation, colocalization of the 2 proteins within GFP-mCherry-positive inclusions was observed, suggesting that LIPA-α-syn recruits and coaggregates with monomeric α-syn (Fig 3D). High magnification confocal images revealed that LIPA-α-syn forms the core of aggregates surrounded by the recruited α-syn-GFP, thus mimicking the characteristics of the α-syn seeding process in α-synucleinopathies [2,26] (S7F Fig, S5 Video). α-Syn-GFP overexpressed alone did not trigger the formation of aggregates in the presence or absence of light stimulation, and no seeding effect was observed when α-syn-GFP was overexpressed with LIPA-Empty or LIPA-α-syn$^{\Delta NAC}$ constructs (S7G Fig). This result confirms that the presence of α-syn in the LIPA construct is required for the seeding of α-syn-GFP. Moreover, we biochemically confirmed LIPA-α-syn seeding capacity using a pull-down assay, and our results revealed that α-syn-GFP co-immunoprecipitated predominantly with LIPA-α-syn inclusions, whereas only a weak α-syn-GFP signal was detected under unstimulated conditions (Fig 3E, S1 Raw images). No α-syn-GFP signal was observed after LIPA-Empty immunoprecipitation with or without blue light stimulation, indicating that the recruitment of monomeric α-syn is specific to LIPA-α-syn and may indicate an α-syn/α-syn homo-seeding process (Fig 3E, S1 Raw images).

To confirm the seeding capacity of LIPA-α-syn aggregates, we tested the ability of purified LIPA-α-syn added to cell culture medium to enter in recipient cells and trigger the aggregation of monomeric α-syn. Confocal imaging of N2a cells overexpressing α-syn-GFP and exposed to purified LIPA-α-syn aggregates revealed that LIPA-α-syn aggregates were present in the cytosol of recipient cells and colocalized with α-syn-GFP (Fig 3F and 3G). These results confirm the capacity of purified LIPA-α-syn inclusions to integrate into recipient cells and recruit the endogenous monomeric counterparts. Similar results were observed in 2 cell lines (N2a and HEK-293T) when using α-syn preformed fibrils (Pffs), whose seeding capacity in cell culture has been previously described [32] (S7H and S7I Fig). α-Syn Pffs were also able to seed the aggregation of LIPA-α-syn in the absence of blue light, but not LIPA-Empty or LIPA-α-syn$^{\Delta NAC}$ (S7J Fig), thus emphasizing the important role of α-syn in the seeding process.

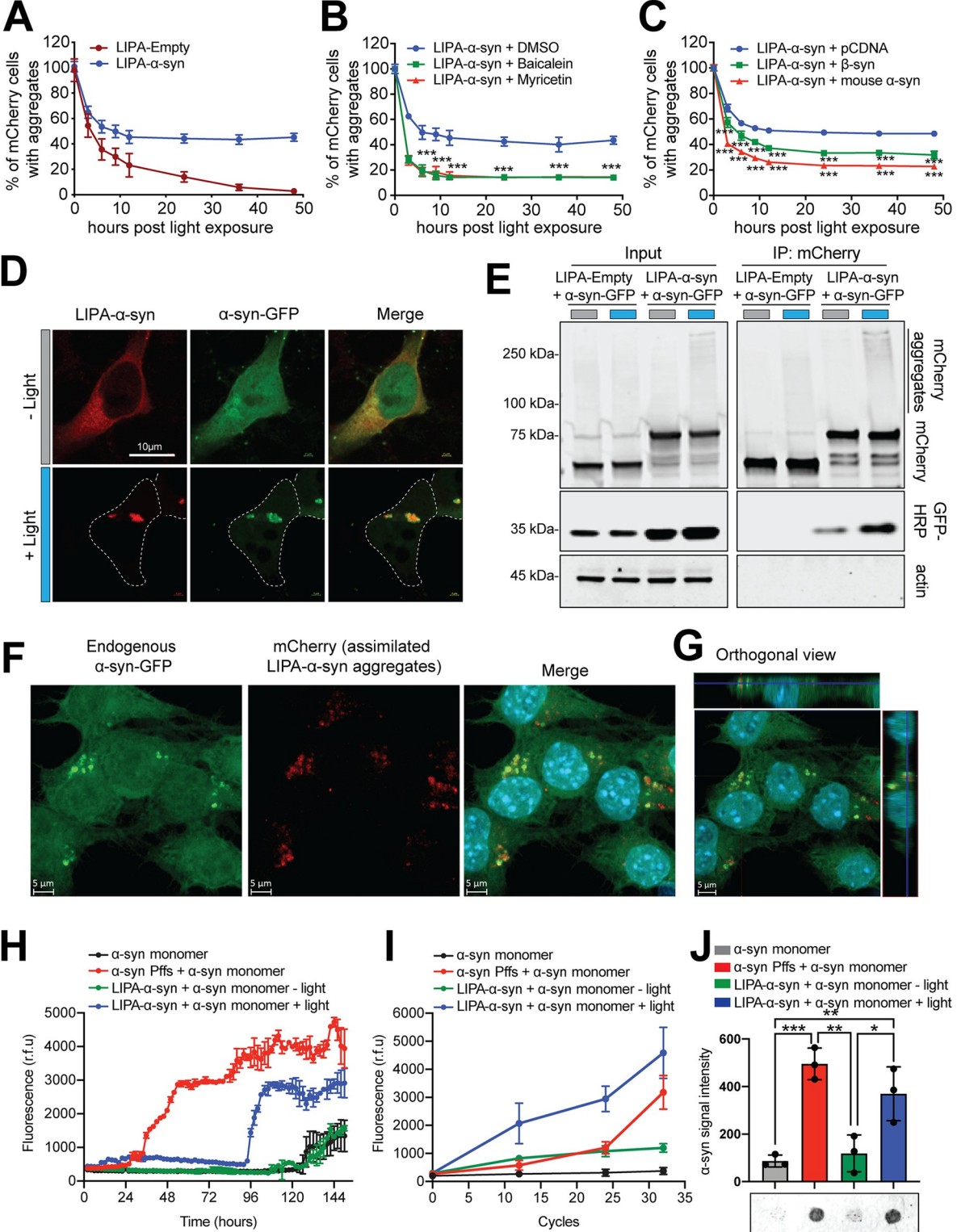

**Fig 3. The LIPA system initiates stable and self-perpetuating α-syn aggregates through the seeding effect.** (**A**) Time course of LIPA-Empty and LIPA-α-syn aggregate dissociation after 12 hours of preexposure to blue light. The data are presented as the percentage of mCherry-positive cells exhibiting inclusions over the total number of mCherry-positive cells normalized to the number of mCherry-positive cells exhibiting inclusions at time zero, which is set as 100% (*n* = 5). (**B**) Impact of treatment with baicalein and myricetin, small-molecule inhibitors of α-syn aggregation, on LIPA-α-syn aggregate stability (*n* = 4). (**C**) Effect of β-syn and mα-syn overexpression on

LIPA-α-syn aggregate stability (*n* = 3). The data are presented as percentage of mCherry-positive cells with inclusions over the total number of mCherry-expressing cells, which was then normalized as 100% at time zero. The data are presented as the means ± SEM. *** *p* ≤ 0.001. (**D**) Confocal microscopy images of HEK-293T cells illustrating the seeding capacity of LIPA-α-syn inclusions in recruiting α-syn-GFP after 12 hours of blue light stimulation (*n* = 3) (scale bar = 10 μm). (**E**) Co-IP of LIPA-α-syn aggregates and α-syn-GFP showing that LIPA-α-syn aggregates interact with and recruit their soluble monomeric counterparts (*n* = 3). (**F**) Confocal representative images and (**G**) orthogonal view showing the seeding capacity of purified LIPA-α-syn added to the culture medium and internalized by recipient N2a cells (*n* = 3) (scale bar = 5 μm). (**H**) RT-QuIC analysis illustrating the kinetics of recombinant α-syn aggregation in the presence of purified LIPA-α-syn aggregates (+light), monomeric LIPA-α-syn (−light), and recombinant α-syn Pffs. The average ThT fluorescence intensity was plotted against time (*n* = 3). The data are presented as the means ± SEM. (**I**) ThT binding kinetics of recombinant α-syn in the presence of purified LIPA-α-syn aggregates (+light), monomeric LIPA-α-syn (−light), and recombinant α-syn Pffs after 32 cycles of PMCA (*n* = 3). (**J**) Filter retardation assay and quantification of α-syn protein levels showing the accumulation of α-syn aggregates in the presence of LIPA-α-syn aggregates and recombinant α-syn Pffs after 32 cycles of PMCA (*n* = 3). The data are presented as the means ± SEM. * *p* ≤ 0.05, ** *p* ≤ 0.01, and **** *p* ≤ 0.0001. The underlying data for (**A**), (**B**), (**C**), (**H**), (**I**), and (**J**) can be found in S1 Data. α-syn, α-synuclein; β-syn, β-synuclein; IP, immunoprecipitation; LIPA, light-inducible protein aggregation; mα-syn, mouse α-syn; PMCA, protein misfolding cyclic amplification; RT-QuIC, real-time quaking-induced conversion; ThT, thioflavin T.

Finally, we assessed the seeding capacity of purified LIPA-α-syn aggregates in vitro using protein misfolding cyclic amplification (PMCA) and real-time quaking-induced conversion (RT-QuIC), 2 highly sensitive assays that allow for the detection of α-syn seeding capacity in biological fluids [33,34]. α-Syn aggregation was monitored by the fluorescence signal emitted by thioflavin T (ThT), a dye that specifically binds to amyloid fibrils. In the 2 assays, we observed that the maximum ThT signal was highest in samples containing LIPA-α-syn aggregates (+light) and in positive control samples (α-syn Pffs) compared to the ThT signal in LIPA-α-syn monomers (−light) or recombinant α-syn monomers alone (negative control) (Fig 3H and 3I, S1 Data). These results confirm the capacity of LIPA-α-syn to seed the aggregation of recombinant monomeric α-syn in vitro. Using a filter retardation assay, we validated that the increase in ThT signal in the presence of LIPA-α-syn aggregates and α-syn Pffs indeed reflects an enhanced accumulation of misfolded α-syn (Fig 3J, S1 Data, S1 Raw images). Notably, neither LIPA-α-syn^ΔNAC nor LIPA-Empty, with or without blue light stimulation, was able to seed recombinant α-syn aggregates in vitro (S7K Fig, S1 Data). Collectively, our results demonstrate that LIPA-induced α-syn inclusions can autoperpetuate through the recruitment and seeding of soluble α-syn monomers, thus mimicking a cardinal pathological hallmark of authentic LBs.

## The LIPA system induces LB-like inclusion formation in vivo and triggers dopaminergic neuronal loss and PD-like motor impairments

To further extend the application of the LIPA system in vivo, we first assessed the capacity of LIPA-α-syn to induce LB-like inclusion formation in the mouse brain. To this end, we used an adeno-associated virus (AAV) delivery system to overexpress LIPA-α-syn directly in the SNc of wild-type (WT) mice. During this surgical intervention, we also implanted a wireless optogenetic device to deliver light to the midbrain. Fifteen days postinjection, LIPA-α-syn aggregation was induced by blue light stimulation for 1 hour every other day over a period of 7 days (Fig 4A) using nontoxic optogenetic stimulation parameters (S8A and S8B Fig, S1 Data). Postmortem analyses showed that AAV2/6 encoded LIPA construct exhibited neuronal tropism, with more than 98% of mCherry cells staining positive for the the neuronal marker, NeuN (S8C and S8D Fig, S1 Data). Moreover, more than 80% of the dopaminergic neurons (TH + cells) expressed the LIPA constructs (S8E and S8F Fig, S1 Data). When exposed to light stimulation, 80% of the TH+-mCherry+ exhibited pS129-positive aggregates (S8E and S8G Fig, S1 Data). Analysis of the midbrain dopaminergic neurons exposed to blue light revealed the presence of mCherry inclusions positive for α-syn, pS129, ubiquitin, p62, and HSP70 and comprised β-pleated sheet content positive for ThS staining, thus recapitulating some of the

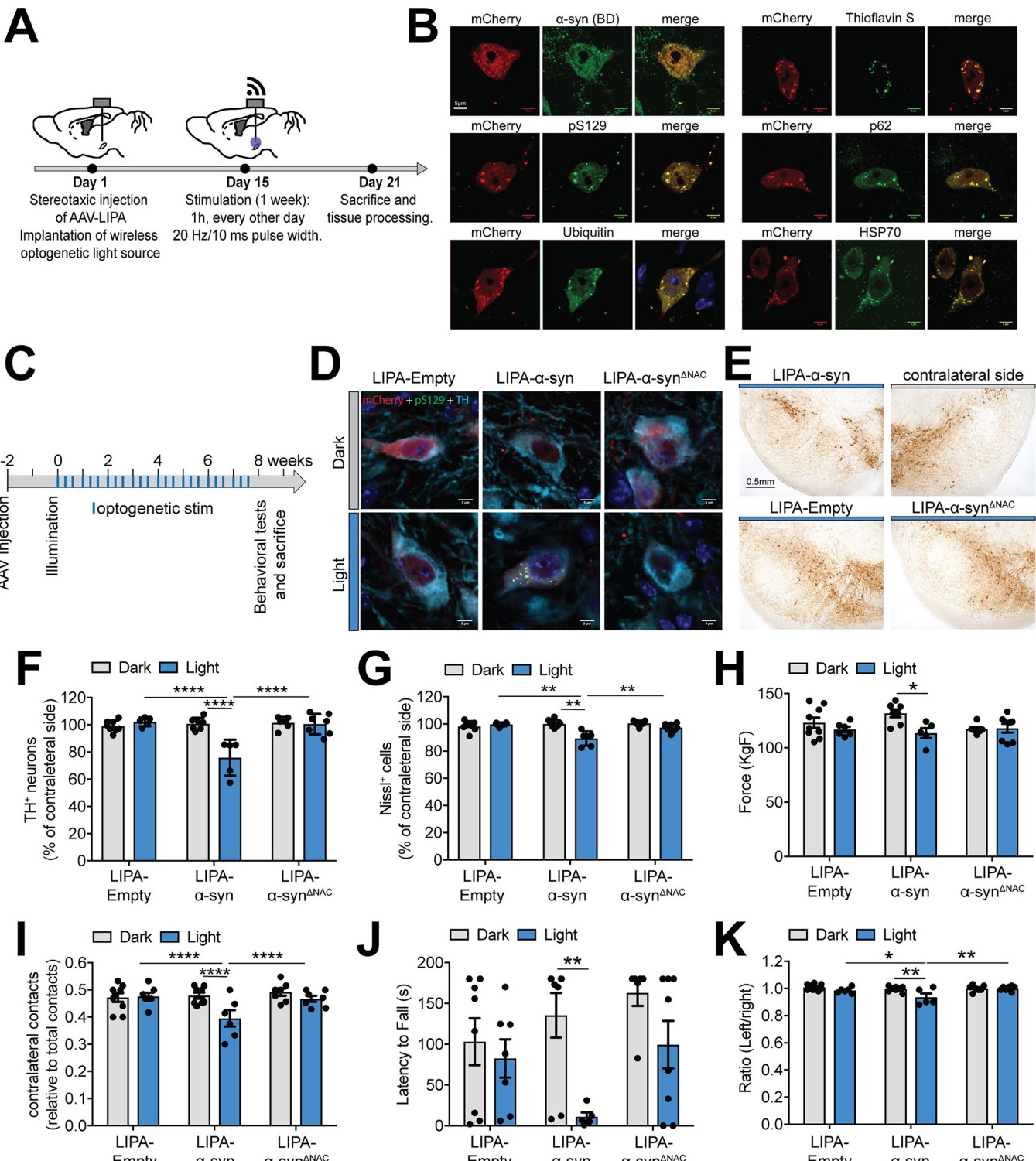

**Fig 4. LIPA-α-syn inclusions recapitulate authentic LB features in vivo and precipitate dopaminergic neuronal loss and parkinsonian-like motor impairments.** (**A**) Experimental design of the experiments using the overexpression and induction of LIPA-α-syn aggregation in the midbrains of WT mice. (**B**) Confocal microscopy images of representative midbrain neurons with LIPA-α-syn aggregates exhibiting authentic LB markers: α-syn (BDlab), pS129, ThS, ubiquitin, HSP70, and p62 ($n$ = 4 mice) (scale bar = 5 μm). (**C**) Experimental design of the long-term impact of LIPA-α-syn aggregation on DA neuronal integrity. Treatment with light stimulation (blue lines) was started 2 weeks post-AAV injection, and blue light was applied for 1 hour every other day for 8 weeks. (**D**) Confocal microscopy images illustrating the LIPA-α-syn or LIPA-Empty inclusions within dopaminergic midbrain neurons. The presence of

pathological (pS129-positive) α-syn aggregates was observed only in dopaminergic neurons overexpressing LIPA-α-syn stimulated with blue light ($n = 4$ mice) (scale bar = 5 μm). (**E**) Representative confocal microscopy images illustrating dopaminergic neuronal loss in the midbrain of mice overexpressing LIPA constructs and exposed to blue light stimulation ($n = 4$ mice) (scale bar = 0.5 mm). (**F**) Stereological quantification of TH-positive dopaminergic neurons and (**G**) total neuronal markers (Nissl) in the midbrains of mice overexpressing LIPA constructs and exposed (or not exposed) to blue light stimulation. The results are expressed as percentage of the contralateral noninjected side ($n = 5$–8 mice per experimental condition). The data are presented as the means ± SEM. * $p \leq 0.05$, ** $p \leq 0.01$, and **** $p \leq 0.0001$. Assessment of the behavioral impairment induced by LIPA constructs overexpression with and without blue light stimulation using (**H**) a grip strength test, (**I**) cylinder test, (**J**) rotarod test, and (**K**) gait test ($n = 5$–9 mice per experimental condition). The data are presented as the means ± SEM. * $p \leq 0.05$, ** $p \leq 0.01$, and *** $p \leq 0.0001$. The underlying data for (**F**) to (**K**) can be found in S1 Data. AAV, adeno-associated virus; α-syn, α-synuclein; LB, Lewy bodies; LIPA, light-inducible protein aggregation; pS129, phosphorylated α-syn at S129; ThS, thioflavin S; WT, wild-type.

cardinal biochemical features of authentic LBs (Fig 4B). In contrast, nonstimulated midbrain neurons were completely devoid of such inclusions (S8H Fig). In a distinct experiment, we were able to overexpress LIPA-α-syn and stimulate the induction of LIPA-α-syn inclusion formation in another brain region, namely, the striatum, thus emphasizing the spatial versatility of the LIPA system (S9 Fig).

Next, we investigated the impact of long-term LIPA-α-syn aggregation on DA neuronal integrity in mice optogenetically stimulated for 1 hour every other day over a period of 8 weeks (Fig 4C). Under control conditions, when no blue light was delivered in the SNc, no mCherry-positive inclusions were observed, whereas optogenetic stimulation induced pathologic pS129-positive inclusions in DA neurons overexpressing LIPA-α-syn, but not in neurons expressing LIPA-α-syn$^{\Delta NAC}$ or LIPA-Empty (Fig 4D). Using unbiased stereological quantification, we observed that the induction of LIPA-α-syn aggregation caused significant TH-positive neuronal loss, while no neuronal death was detected in nonstimulated midbrains or in stimulated midbrains overexpressing LIPA-Empty or LIPA-α-syn$^{\Delta NAC}$ (Fig 4E and 4F, S1 Data). Nissl-positive neuronal quantification confirmed that TH-positive neuronal loss was indeed due to neurodegeneration and not the loss of the DA phenotype (Fig 4G, S1 Data). Interestingly, LIPA-α-syn-induced DA neurodegeneration was accompanied by significant parkinsonian-like motor deficits, including reduced grip strength (Fig 4H, S1 Data), asymmetry in the use of contralateral forepaw in the cylinder test (Fig 4I, S1 Data), a reduction in coordination (Fig 4J, S1 Data), and gait abnormalities (Fig 4K, S1 Data). Collectively, our results demonstrate that the LIPA system allows for the induction of PD-related neuropathology and behavioral impairment in vivo, thus offering a viable rodent model of α-syn aggregation to study PD and related synucleinopathies.

## LIPA-induced α-syn inclusions disrupt nigrostriatal transmission

The spatiotemporal control of LIPA-α-syn gives an unprecedented opportunity to study early hallmarks of altered nigrostriatal transmission following α-syn aggregation. Previous studies reported that a certain rate and synchronicity of striatal neuron activity is required for proper motor movements and that alterations in DA transmission induce changes in the cellular and circuitry properties of the striatum [35–37]. However, the changes in striatal neurons at the cellular and network levels during the early stages of α-syn aggregation remain unclear. To address this question, we used Ca$^{2+}$ imaging to record the activity of striatal neurons in freely moving mice through a double opto-endoscopic system allowing for optogenetic stimulation of midbrain neurons overexpressing LIPA-α-syn and endoscopic imaging of striatal neurons expressing a genetically encoded Ca$^{2+}$ indicator, GCaMP6s (Fig 5A and 5B). We first imaged the activity of striatal neurons under baseline conditions every other day for 4 days. Then, we optogenetically stimulated LIPA-α-syn aggregation in the midbrain for 1 hour per day every other day and performed Ca$^{2+}$ imaging in the striatum for a period of 7 days. This procedure was followed by poststimulation imaging of striatal neurons for 10 additional days without

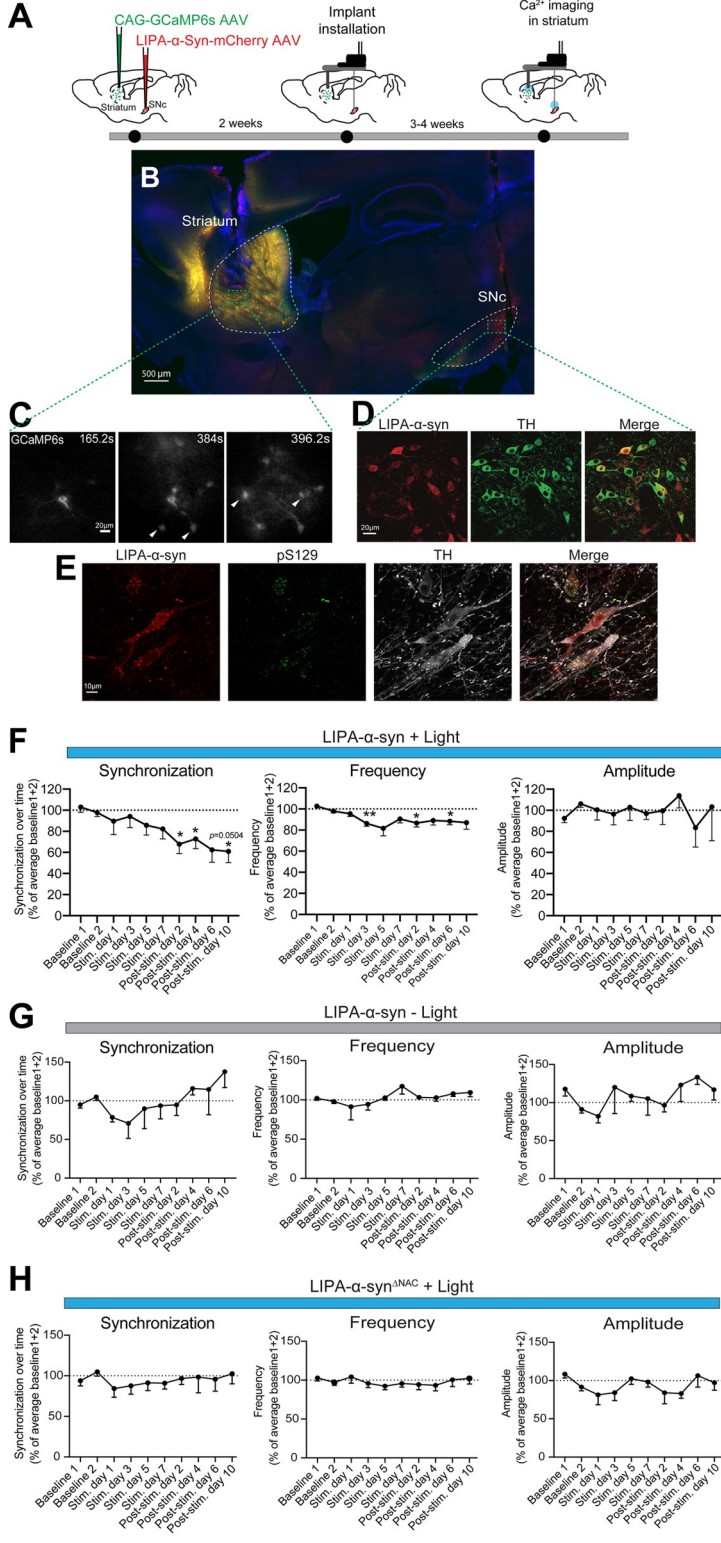

**Fig 5. LIPA-α-syn aggregation disrupts nigrostriatal neuronal transmission.** (**A**) Schematic representation of the in vivo experimental paradigm of light-induced α-syn aggregation in the SNc and Ca²⁺ imaging in the striatum. (**B**) Confocal microscopy image reconstitution of a sagittal brain slice illustrating the expression of LIPA-α-syn in the SNc and the Ca²⁺ indicator GCaMP6s in the striatum (*n* = 5 mice) (scale bar = 500μm). (**C**) Representative time-lapse live imaging illustration of mini-endoscopic Ca²⁺ in the striatum. Arrowheads indicate striatal neurons showing Ca²⁺

activity (scale bar = 20μm). (**D**) Confocal images illustrating the presence of LIPA-α-syn expression in the majority of dopaminergic TH-positive neurons (*n* = 5 mice) (scale bar = 20 μm). (**E**) High magnification of confocal images showing that pathological pS129-positive LIPA-α-syn aggregates persisted in dopaminergic neurons 10 days post-optogenetic stimulation (*n* = 4–6 mice) (scale bar = 10 μm). (**F**) Temporal trajectories of synchronicity, frequency, and amplitude of Ca$^{2+}$ transients before, during, and after light-induced α-syn aggregation (*n* = 4–5 mice). (**G**) Temporal trajectories of synchronicity, frequency, and amplitude of Ca$^{2+}$ transients in the absence of light-induced LIPA-α-syn aggregation (*n* = 3 mice). (**H**) Temporal trajectories of synchronicity, frequency, and amplitude of Ca$^{2+}$ transients before, during, and after light-induction in LIPA-α-syn$^{\Delta NAC}$ (*n* = 4–5 mice). The data are presented as the means ± SEM. * $p \leq 0.05$ and ** $p \leq 0.01$. The underlying data for (**F**) to (**H**) can be found in S1 Data. α-syn, α-synuclein; LIPA, light-inducible protein aggregation; pS129, phosphorylated α-syn at S129; SNc, substantia nigra pars compacta.

midbrain optogenetic stimulation. This experimental paradigm allowed for the monitoring and recording of striatal neuronal activity (Fig 5C). Moreover, postmortem analysis revealed that the majority of the dopaminergic neurons expressed LIPA-α-syn, and the light-induced pS129-positive aggregates were persistent in the dopaminergic neurons 10 days poststimulation (Fig 5D and 5E). We monitored synchronicity, frequency, and amplitude of Ca$^{2+}$ activity within striatal neurons before, during, and after light-induced α-syn aggregation in the midbrain (S6 Video, S10 Fig, S1 Data). Analyses revealed that light-induced α-syn aggregation decreased the synchronization and frequency of Ca2+ activity in the striatal neurons shortly after the onset of stimulation without affecting the amplitude of Ca$^{2+}$ transients (Fig 5F, S1 Data). Interestingly, no effect on the activity of striatal neurons was detected in nonstimulated LIPA-α-syn or in light-stimulated LIPA-α-syn$^{\Delta NAC}$ conditions (Fig 5G and 5H, S1 Data). These observations suggest that α-syn aggregation in the midbrain can rapidly impact striatal neuronal activity, reflecting the early hallmarks of altered nigrostriatal dopaminergic transmission following α-syn aggregation. Both the synchronization and frequency of the impairment induced by LIPA-α-syn aggregation continued to be exacerbated for several days (at least 10 days) after optogenetic stimulation was terminated (Fig 5F, S1 Data). These observations suggest that the effects of LIPA-α-syn aggregation, which persisted for several days poststimulation on nigrostriatal dopaminergic transmission, may be sustained for a long period after inclusion formation, reflecting in some ways the disease progression. Collectively, our data demonstrate that the LIPA system offers an innovative tool for monitoring and dissecting cellular events related to α-syn pathology in freely moving animals.

## Discussion

Building on the principles of optogenetics, we developed a novel system capable of triggering the formation of authentic LBs in cell culture as well as in vivo. We showed that this system can be used to study and monitor not only different aspects of α-syn aggregation (e.g., the seeding effect, autoperpetuation, and stability) but its consequences on the activity and survival of midbrain neurons, as well as on the development of motor deficits. To our knowledge, this is a unique system for controlling and monitoring PD-related processes in real time with high spatiotemporal resolution.

### LIPA-α-syn inclusions recapitulate several cardinal features of authentic LBs in a rapid and reproducible manner

In the present study, we reported that light-induced α-syn aggregates exhibited several features of authentic human LBs, including biochemical modifications (increased phosphorylation and ubiquitination) [38], β-sheet structures [17], and p62- and HSP70-positive signals [22]. At the ultrastructural level, the LIPA-α-syn inclusions consisted of crowded lipid membranous material, comprising vesicles and dysmorphic organelles (mitochondria, the Golgi apparatus, and

the endoplasmic reticulum) and fibrillar structures, faithfully mimicking the ultrastructure of authentic LBs recently described using CLEM [23]. Although the α-syn fibrils were detected in purified LIPA-α-syn aggregates, no such structures were observed LIPA-α-syn inclusions in fixed HEK-293T cells. This may have been the result of technical limitations, such as small size of α-syn fibrils and the crowdedness of the inclusions, without excluding the possibility that a longer maturation time may be required for the observation of typical α-syn fibrillar forms. Alternatively, special staining combined to a longer maturation time might be required for the proper visualization of these fibrils.

In addition to the biochemical and ultrastructural characteristics observed, the LIPA-α-syn inclusions exhibited a key functional feature of authentic LBs: the seeding capacity [2,26]. During the seeding process, LIPA-α-syn aggregates depicted the capacity to self-perpetuate by recruiting endogenous monomeric α-syn after single-dose and transient light stimulation. Taken together, these observations demonstrate that our model recapitulates the majority of the cardinal features of LB pathology, thus offering a unique tool for studying the biology of protein inclusions and their induction and formation processes.

## Benefits of the precise temporal and spatial control of LIPA-induced α-syn aggregation

The use of the LIPA system allows the control of α-syn aggregation and LB formation at remarkable temporal resolution. First, the LIPA system allows light induction with fast kinetics. In the present study, we observed α-syn inclusion formation within a few seconds after light stimulation, which was sustained for several hours. This precise temporal control of α-syn aggregation makes the LIPA system a unique tool for investigating rapid cell responses to acute α-syn aggregation, i.e., synaptic transmission and plasticity. Second, the LIPA system enabled the control of experimental intervention for relatively longer periods (48 hours in cell culture and several weeks in vivo), allowing us to study the long-term neuropathological effects of these inclusions in living cells and in the central nervous system (CNS). Notably, the LIPA system is extremely sensitive to light, and the use of very low blue light pulse intensities was sufficient to induce rapid and robust α-syn aggregation, thus minimizing any potential cell toxicity related to blue light stimulation per se.

Spatial subcellular control of α-syn aggregation using optogenetics represents another major advantage of the LIPA system. In this study, we demonstrated the effectiveness of our model in controlling protein aggregation at the single-cell level, with the possibility of inducing α-syn aggregation in a specific cell subcompartment. This concept of subcellular compartmentation seems to be very relevant for the LIPA system, as live imaging data suggest that LIPA-Empty aggregation may occur first in the nucleus and subsequently extend to the cytosol (S2 Video). This observation suggests that LIPA aggregation can behave differently depending on where the process is taking place and based on the expression levels of the LIPA constructs in different subcellular compartments. In vivo, the combination of optogenetic and AAV delivery systems allowed the expression and induction of α-syn aggregation in specific cerebral regions, as we reported here, in 2 basal ganglia nuclei, namely, the SNc and the striatum, both main areas of degeneration in PD. This in vivo versatility allows to study protein aggregation in a tissue-specific manner and to address questions related to differential neuronal vulnerabilities within the CNS.

## LIPA-induced α-syn aggregates precipitate dopaminergic neuronal loss and induce parkinsonian-like symptoms

Although the presence of LBs in vulnerable neuronal populations has been established as the cardinal neuropathological hallmark of PD and related α-synucleinopathies, the precise impact

of these inclusions on DA neuronal integrity remains highly debated. While some studies have proposed a detrimental role of LBs on DA neurons, showing increased DA neuronal vulnerability [39,40], other clinical studies reported incidental LBs in 5% to 12% of neurologically healthy subjects, suggesting that LBs are not always toxic and may form during normal aging [41,42]. A recent hypothesis even suggested that LBs may play a neuroprotective role by trapping and promoting the elimination of nondegraded toxic proteins and organelles [43–45]. In the present study, we reported that α-syn aggregation was associated with a significant loss of DA neurons in the midbrain and the manifestation of motor behavior impairments. These observations were in line with the toxic effect of α-syn inclusions on DA neurons observed in vivo. Moreover, our observations also suggested that the aggregated form, including soluble oligomers, rather than monomeric α-syn accumulation, imposed the most toxic effect. This supposition was supported by experimental evidence showing that the α-syn oligomeric species formed during the aggregation process, not α-syn overexpression, had deleterious effects on the neuronal homeostasis [46–48]. Notably, most of the behavioral parameters tested were mildly impaired, except for motor coordination, which showed dramatic impairment, as indicated in the rotarod test. Given that this test is very sensitive to striatal neuronal dysfunction caused by disruption of DA transmission [49,50], the marked effect observed in the present study could have been due to a profound alteration of nigrostriatal dopaminergic transmission, as detected very early in our LIPA in vivo model using $Ca^{2+}$ imaging.

## The LIPA system helps monitor early physiological impairment associated with α-syn aggregation: Disruption of nigrostriatal dopaminergic transmission

To date, most assessments of basal ganglia pathophysiology have been performed at the end stages of DA loss, and, therefore, very little is known about functional changes occurring during the initial stages of DA decrease when motor symptoms have not yet manifested [51–54]. One of the advantages of the LIPA system is the possibility of combining optogenetic control of α-syn aggregation with live imaging approaches in vivo. This combination allowed us to perform time-lapse monitoring of functional responses in the striatal neurons immediately following α-syn aggregation in the midbrain, an approach otherwise impossible to perform using current experimental models of PD. At the cellular level, we observed a decreased frequency of $Ca^{2+}$ transients and changes in the synchronous activity of postsynaptic striatal neurons. Changes in the rate of firing of basal ganglia output neurons have been previously reported during the late stages of disease progression [35,36], although the magnitude and sign of these changes vary across studies [55,56]. Altogether, our cellular and network analyses of postsynaptic striatal neurons indicate a temporal trajectory of nigrostriatal dopaminergic transmission alterations following α-syn aggregation.

## Limitations of the LIPA-induced α-syn aggregation model

The main limitation of the LIPA system is the large size of the construct, where α-syn is fused to CRY2olig and mCherry proteins. Despite the fact that one may wonder if the fusion construct could affect α-syn aggregation properties, our results clearly demonstrated that α-syn not only continued to exhibit aggregation and seeding capacity but also exhibited aggregation patterns and kinetics on the CRY2-mCherry fusion construct. This conclusion was based on the following experimental evidence: (1) the presence of α-syn accelerated light-induced LIPA aggregation, whereas the presence of the nonaggregatable form of $\alpha\text{-syn}^{\Delta NAC}$ ($\alpha\text{-syn}^{\Delta NAC}$) precluded inclusion formation; (2) transient light stimulation was sufficient to induce and maintain LIPA-α-syn aggregates for several days, whereas LIPA-Empty aggregates disappeared

within a few hours after light stimulation was terminated; and (3) LIPA-induced α-syn inclusions were able to seed the aggregation of monomeric α-syn, leading to aggregation autoperpetuation. This phenomenon is specific to aggregating proteins, such as α-syn, and was not observed with the LIPA-Empty construct. Collectively, these observations demonstrate that the size of the LIPA constructs does not affect α-syn behavior, including its aggregation process in living cells, thus supporting the validity of this model for studying protein aggregation and LB formation.

## Conclusions

Together, our data demonstrate that the LIPA system can reliably induce α-syn aggregation with high spatial and temporal resolution, offering a unique opportunity to address unresolved questions, i.e., those related to α-syn inclusion propagation between subcellular compartments, α-syn propagation between cells and between brain regions, as well as questions related to differential vulnerability between various neuronal types. Importantly, a similar approach has been successfully applied to other proteinopathy-related proteins, including TDP-43 [57,58] and amyloid precursor protein (APP) [59].

## Materials and methods

### Plasmid construction and production of recombinant adeno-associated 2/6 viral vectors

CRY2olig-mCherry plasmid was kindly provided by Dr. Chandra Tucker (Addgene plasmid # 60032). pcDNA-human α-syn plasmid was kindly provided by Dr. Hilal Lashuel (EPFL, Switzerland). pcDNA-human α-syn$^{\Delta NAC}$, missing the NAC region (aa71-82), plasmid was kindly provided by Dr. Benoit Giasson (University of Florida, USA). pAAV-CMV-MCS plasmid was kindly provided by Dr. Bernard Schneider (EPFL, Switzerland). α-syn mCherry plasmid was kindly provided by Dr. Sang Myun Park (Ajou University, South Korea).

To generate LIPA constructs, the cDNA encoding human α-syn or human α-syn$^{\Delta NAC}$ were subcloned in CRY2olig-mCherry and verified by sequencing. To generate plasmids for the production of adeno-associated viral vectors, the cDNAs encoding LIPA-α-syn (α-syn-CRY2olig-mCherry) was subcloned in the pAAV-CMV-MCS shuttle plasmid, using standard cloning procedures, and verified by sequencing.

The production of the recombinant pseudotyped AAV2/6 vectors (serotype 2 genome/serotype 6 capsid) and relative infectivity titers were performed by the Canadian Neurophotonics Platform (CERVO, Quebec City). The final viral titers were as follows: $1.5 \times 10^{13}$ GC/ml for AAV-LIPA-Empty, $1 \times 10^{13}$ GC/ml for AAV-LIPA-α-syn, and $7 \times 10^{12}$ GC/ml for AAV-LIPA-α-syn$^{\Delta NAC}$.

### Cell culture and DNA transient transfection

Human HEK-293T and N2a cells (ATCC) were maintained in high-glucose Dulbecco's modified Eagle's medium (DMEM) (D5796; Sigma-Aldrich, St. Louis, MO, USA) supplemented with 10% fetal bovine serum (FBS) (F1051; Sigma-Aldrich, St. Louis, MO, USA) and 1% penicillin/streptomycin (15-140-122; ThermoFisher Scientific/Gibco, Waltham, MA, USA) at 37°C and 5% $CO_2$. Cells were transfected using calcium phosphate, FastFect transfection (9K-010-0001; Feldan, Quebec, QC, Canada) or Lipofectamine 2000 (11668027; ThermoFisher Scientific, Waltham, MA, USA) according to a standard protocol, leading to a transfection efficiency of more than 95%. The amount of DNA used for each plasmid, 0.5 μg of LIPA-Empty, 0.6 μg LIPA-α-syn, and 1 μg LIPA-α-syn$^{\Delta NAC}$ per well in a 6-well plate was normalized for equal

mCherry expression as measured by western blot. Cells were also transfected with α-syn-mCherry at a concentration of 1 μg per well in a 6-well plate.

## Live cell imaging and 3D animation

HEK-293T cells were seeded on 35 mm MatTek Poly-D-Lysine coated glass bottom dishes, 24 hours postplating, cells were transiently transfected at 30% to 50% confluence using Lipofecta-mine 2000, with 0.5 μg of LIPA-Empty, 0.6 μg LIPA-α-syn, and 1 μg LIPA-α-syn$^{\Delta NAC}$. Twenty-four hours posttransfection, live imaging was performed using an Olympus IX-81 FV1000 confocal microscope equipped with a stage top incubator (Leica Biosystems, Concord, ON, Canada). LIPA photoactivation was induced by laser illumination (488 nm) at 30% (15.5 μW/mm$^2$). Time-lapse acquisitions were performed every 2 minutes, photoactivation of a small ROI in the high-magnification images (60×) or the whole field in the low-magnification images (20×) were performed with the 488-nm laser wavelength just prior to each frame recording. During the entire recording time, conditions were controlled, and cells were maintained at 37°C, 5% $CO_2$. Time-lapse microscopy data were analyzed using the Imaris software version 7.6.1 (Bitplane, Zurich, Switzerland). Images were segmented using the isocontour feature in the Surpass module. Highly expressing mCherry-positive aggregates were detected based on the local contrast, their shape, and volume. Cell volume was detected as low-intensity mCherry expression volume and used to normalize the quantity of detected aggregates. Animation rendering was performed with the Imaris software, and videos were exported using QuickTime Pro 7 (Apple, Cupertino, CA, USA).

## Induction of protein aggregation, immunocytochemistry, and cell quantification in culture

HEK-239T cells were seeded on 0.2% gelatin-coated glass coverslips in 24-well plates at a density of $10^4$ cells per well (50% confluence). Twenty-four hours postplating, cells were transiently transfected using calcium phosphate protocol, with 0.5 μg of LIPA-Empty, 0.6 μg LIPA-α-syn, and 1 μg LIPA-α-syn$^{\Delta NAC}$ per 4 wells, to obtain similar protein expression levels in all conditions. Twenty-four hours posttransfection, cells were exposed to blue light (λ = 456 nm) using UHP-T-DI-LED series Ultra High-Power LEDs (Prizmatix, Southfield, MI, USA), at the intensity of 0.8 mW/mm$^2$ or as otherwise indicated, measured using LPM-100 light power meter (Amuza, San Diego, CA, USA). Cells were collected at different time points (as indicated in each figure) and fixed in 4% paraformaldehyde (PFA) + 3% sucrose for 15 minutes at room temperature (RT), then washed 3 times (5 minutes) with PBS and permeabilized with a solution containing (0.25% Triton X-100 in PBS) for 30 minutes at RT. Cells were incubated in a blocking buffer solution containing (5% BSA, 0.1% Triton X-100 in PBS) for 1 hour at RT. Coverslips were incubated with primary antibodies (see Table 1) diluted in blocking buffer solution for 2 hours at RT. Coverslips were washed 3 times in blocking buffer solution (10 minutes), and appropriate Alexa Fluor secondary antibodies (see Table 1) were applied for 1 hour at RT. Coverslips were washed twice with PBS (10 minutes) and counterstained with the fluorescent nuclear stain diluted in 4,6-diamidino-2-phenylindole (DAPI) (D3571; Sigma-Aldrich, St. Louis, MO, USA) (1:50,000) for 5 minutes. Coverslips were then washed twice with PBS (10 minutes) and then mounted on Prolong Gold Antifade (P36975; Molecular Probes, Eugene, OR, USA). For ThS staining (T1892; Sigma-Aldrich, St. Louis, MO, USA), fixed and permeabilized cells were incubated with 0.05% ThS for 15 minutes and were washed 3 times (10 minutes) with 70% ethanol, then cells were directly counterstained with DAPI. All immunocytochemistry experiments were performed with gentle shaking protected from light. Cells were imaged using a Zeiss LSM800 confocal microscope (Zeiss, Oberkochen, Germany).

**Table 1. List of antibodies used in the present study.**

| Antigen/species | Antibody name/catalog number | Epitope | Concentration immunoblotting | Concentration IHC or ICC | Source |
|---|---|---|---|---|---|
| Primary antibodies | | | | | |
| α-syn/mouse | Syn1 / 610787 | 15–123 | 1:1,000 | 1:1,000 | BD Laboratory (Billerica, MA, USA) |
| Human α-syn/ mouse | α-synuclein (211) / sc-12767 | 121–125 | | 1:1,000 | Santa Cruz Biotech (Dallas, TX, USA) |
| α-syn/ rabbit | Syn FL-140 / sc-10717 | 65–91 | | 1:1,000 | Santa Cruz Biotech (Dallas, TX, USA) |
| mCherry/ rabbit | Anti-mCherry antibody/ AB167453 | mCherry | 1:1,000 | | Abcam (Cambridge, UK) |
| pS129 α-Syn/ mouse | WAKO / pSyn #64 | Phospho-Ser129 | | 1:2,000 | WAKO (Richmond, VA, USA) |
| Ubiquitin/ rabbit | Ubiquitin / Z0458 | | | 1:1,000 | Dako (Santa Clara, CA, USA) |
| Ubiquitin/ rabbit | Anti-Ubiquitin antibody/ AB7780 | Recombinant full-length protein corresponding to Human Ubiquitin | | 1: 1,000 | Abcam (Cambridge, UK) |
| p62/ mouse | SQSTM1(D-3) / sc-28359 | 151–440 | | 1:1,000 | Santa Cruz Biotech (Dallas, TX, USA) |
| HSP70/ rabbit | HSP 70/HSC 70 (H-300) sc-33575 | 342–641 | | 1:1,000 | Santa Cruz Biotech (Dallas, TX, USA) |
| HSP70/ mouse | Anti-Hsp70 antibody [5A5]/ AB2787 | N terminal-Hsp70 aa 122–264 | | 1:100 | AbCam (Cambridge, UK) |
| CD9/ rabbit | CD9 Recombinant Monoclonal Antibody (SA35-08)/ MA5-31980 | C terminal Human CD9 | | 1:50 | Invitrogen (Waltham, MA, USA) |
| EEA1/ rabbit | EEA1 (C45B10) mAb/3288 | Residues surrounding Ser70 of human EEA1 protein | | 1:250 | Cell Signaling tech (Danvers, MA, USA) |
| Lamp1/ rabbit | Anti-LAMP1 antibody— Lysosome Marker/ab24170 | Human LAMP1 aa 400 to the C-terminus (C terminal) conjugated to keyhole limpet haemocyanin | | 1:200 | Abcam (Cambridge, UK) |
| Lamp2/ mouse | LAMP2 Monoclonal Antibody (H4B4)/MA1-205 | | | 1:200 | ThermoFisher Scientific (Waltham, MA, USA) |
| MAP2/ rabbit | MAP2 Antibody /4542 | all isoforms of MAP2 total protein | | 1:1,000 | Cell Signaling tech (Danvers, MA, USA) |
| MAP2/ mouse | Monoclonal Anti-MAP2 (2a +2b) clone AP-20 / M1406 | | | 1:1,000 | Sigma-Aldrich (St. Louis, MO, USA) |
| NeuN/ mouse | Anti-NeuN Antibody, clone A60 / MAB377 | | | 1:1,000 | Millipore (Temecula, CA, USA) |
| TH/ mouse | Anti-Tyrosine Hydroxylase Antibody, clone LNC1 / MAB318 | Recognizes an epitope on the outside of the regulatory N-terminus | | 1:1,000 | Millipore (Temecula, CA, USA) |
| Beta-actin/ mouse | β-actin clone BA3R / G043 | Beta-actin N-terminal peptide-KLH conjugates. | 1:10,000 | | Abm (Vancouver, BC, Canada) |
| Secondary antibodies | | | | | |
| IRDye 680RD Goat anti-Rabbit IgG Secondary Antibody | 680RD-conjugated goat anti-rabbit/ 926–68071 | | 1:20,000 | | LI-COR Biosciences (Lincoln, NE, USA) |
| IRDye 800CW Goat anti-Rabbit IgG Secondary Antibody | 800CW-conjugated goat anti-rabbit/ 926–32211 | | 1:20,000 | | LI-COR Biosciences (Lincoln, NE, USA) |
| IRDye 680RD Goat anti-Mouse IgG Secondary Antibody | 680RD-conjugated goat anti-mouse/ 926–68070 | | 1:20,000 | | LI-COR Biosciences (Lincoln, NE, USA) |

(*Continued*)

**Table 1.** (Continued)

| Antigen/species | Antibody name/catalog number | Epitope | Concentration immunoblotting | Concentration IHC or ICC | Source |
|---|---|---|---|---|---|
| IRDye 800CW Goat anti-Mouse IgG Secondary Antibody | 800CW-conjugated goat anti-mouse/ 926–32210 | | 1:20,000 | | LI-COR Biosciences (Lincoln, NE, USA) |
| Goat Anti-Mouse IgG Antibody (H+L), Biotinylated | Biotinylated Goat Anti-Mouse/ BA-9200 | | | 1:500 | Vector Laboratories (Burlingame, CA, USA) |
| Alexa Fluor 488 goat anti-rabbit (H+L) | Alexa Fluor 488 goat anti-rabbit/ A-11008 | | | 1:1,000 | Invitrogen (Waltham, MA, USA) |
| Alexa Fluor 633 goat anti-rabbit (H+L) | Alexa Fluor 633 goat anti-rabbit/ A21071 | | | 1:1,000 | Invitrogen (Waltham, MA, USA) |
| Alexa Fluor 488 goat anti-mouse (H+L) | Alexa Fluor 488 goat anti-mouse/ A-11029 | | | 1:1,000 | Invitrogen (Waltham, MA, USA) |
| Alexa Fluor 633 goat anti-mouse (H+L) | Alexa Fluor 633 goat anti-mouse/ A-21052 | | | 1:1,000 | Invitrogen (Waltham, MA, USA) |
| Alexa Fluor 680 goat anti-rabbit (H+L) | Alexa Fluor 680 goat anti-rabbit/ A-21109, | | | 1:500 | Invitrogen (Waltham, MA, USA) |
| Alexa Fluor 680 goat anti-mouse (H+L) | Alexa Fluor 680 goat anti-mouse/ A-21058 | | | 1:1,000 | Invitrogen (Waltham, MA, USA) |

The percentage of mCherry cells with positive aggregates (cells with one single to several aggregates were counted as mCherry -cells with aggregates) is calculated as follows: # of cells expressing mCherry-positive aggregates/total number of mCherry-positive cells. For all experiments, a total of 8 to 10 images/condition were collected using a fluorescent microscope (20X objective) (Nikon Eclipse 80i) (Nikon Instruments, Melville, NY, USA), and a total of 220 to 250 cells were quantified by 2 blinded experimenters using the ImageJ software.

Cry2 phenotype rescue experiments were performed as follows: HEK-293T cells were plated in 24-well plate and transfected using the calcium phosphate standard protocol. Briefly, 0.25 μg of LIPA-α-syn$^{\Delta NAC}$, or 0.25 μg of pCIBN(ΔNLS)-pmGFP (Addgene, #26867) or the combination of the 2 plasmids were used. Media was replaced with fresh DMEM media 5 hours posttransfection; cells were exposed or not to the blue light for 2 hours. The cells were then washed and fixed. The percentage of mCherry cells with positive aggregates (cells with one single to several aggregates positive for mCherry) were counted as mCherry positive cells with aggregates. A total of 5 images/condition were collected using a confocal microscope (40X objective Carl Zeiss LSM800), and a total of 150 to 200 cells were quantified for each condition in each experiment ($n = 4$) using the ImageJ software.

### Forward programming of NGN2 hiPSC cells into ineurons and ineuron transfection protocol

We used Ngn2-generated iNeurons from hiPSCs kindly provided from Dr. Thomas M. Durcan's lab. For iNeuron differentiation, we used the protocol published from Sudhof group [60]. Before the start of differentiation, hiPSCs were maintained in mTeSR1 media (mTeSR1 Basal medium and mTeSR1 5X supplement) (85857; Stemcell Technologies, Vancouver, BC, Canada) and seeded on Matrigel-coated plates (Matrigel Matrix hESC-qualified) (354277; ThermoFisher Scientific, Waltham, MA, USA). hiPSCs were split with gentle cell dissociation reagent (07174; Stemcell Technologies, Vancouver, BC, Canada), with every passage of hiPSCs, mTeSR1 media was supplemented with ROCK Inhibitor Y27632 (S1049; Selleck Chemicals, Houston, TX, USA). When starting differentiation, we followed the differentiation protocol

published from Sudhof group [60]. Briefly, hiPSCs were dissociated into single cells using accutase (A1110501; ThermoFisher Scientific, Waltham, MA, USA), seeded on Matrigel-coated plates in mTeSR1 media with ROCK Inhibitor Y27632. One day after seeding, the media was switched to N2 medium (DMEM/F12 (10565042–018; ThermoFisher Scientific/ Gibco, Waltham, MA, USA), N2 (17504–044; ThermoFisher Scientific/Gibco, Waltham, MA, USA), B27 (17502–048; ThermoFisher Scientific/Gibco, Waltham, MA, USA), NEAA (321-011-EL; Wisent Life Sciences, Quebec, QC, Canada), BDNF (450–02; Peprotech, Rocky Hill, NJ, USA), GDNF (450–10; Peprotech, Rocky Hill, NJ, USA), laminin (CC095; Sigma-Aldrich, St. Louis, MO, USA) supplemented with 2 μg/ml doxycycline (631311; Takara Bio Company, Mountain View, CA, USA) to induce NGN2 expression. On Day 2, cells were dissociated with accutase and plated on PO/laminin-coated coverslips, and at this point, the media was switched to Day 2 media (neurobasal medium (21103–049; ThermoFisher Scientific/Life Technologies, Waltham, MA, USA), N2, B27, GlutaMax (35050061; ThermoFisher Scientific, Waltham, MA, USA), NEAA, BDNF, GDNF, laminin supplemented with 2 μg/ml doxycycline. Cells were maintained in Day 2 media using long-term doxycycline induction.

Seven days postdifferentiation, iNeurons were transiently transfected with the LIPA construct (1 μg LIPA-α-syn) using ViaFect (E4981; Promega, Madison, WI, USA) following the manufacturer's guidelines. Briefly, neuronal culture media was replaced with Opti-MEM (31985070; ThermoFisher Scientific/ Gibco, Waltham, MA, USA) half an hour prior to transfection. DNA was combined with the reagent at a 1:4 ratio (DNA: reagent), and the transfection mix was then added to the iNeurons, and media was replaced with neuronal media 4 hours posttransfection. Twenty-four hours posttransfection, media was replaced with fresh Day 2 media, and media was then changed every 2 days. Seventy-two hours posttransfection, iNeurons were exposed to the blue light for 10 hours applying the same parameters as described in (section Induction of protein aggregation, immunocytochemistry, and cell quantification in culture) at the intensity of 0.4 mW/mm2.

After illumination, iNeurons were washed once with PBS, and then fixed with cold 4% PFA + 3% sucrose for 20 minutes at RT. Cells were then washed 3 times (5 minutes) with ice-cold PBS and blocked/permeabilized with buffer containing (0.20% Tween20 and 3% BSA in PBS) for 30 minutes at RT. Coverslips were then incubated O.N at 4°C with primary antibodies α-syn (BD), psyn, P62, HSP70, ubiquitin, MAP2, diluted in blocking buffer (see Table 1). Coverslips were then washed 3 times (5 minutes) in PBS solution, and then Alexa Fluor secondary antibodies (see Table 1) were applied for 2 hours at RT. Coverslips were washed twice with PBS (5 minutes) and counterstained with DAPI at (1:5,000) for 5 minutes. Coverslips were then washed 2 times with PBS (5 minutes) and then mounted using Fluoromount-G. For ThS staining (T1892; Sigma-Aldrich, St. Louis, MO, USA), fixed and permeabilized cells were incubated with 0.05% ThS for 15 minutes and washed 4 times with 70% ethanol for 10 minutes. Cells were then counterstained with DAPI as described above. Cells were imaged using a Zeiss LSM800 confocal microscope (Zeiss, Oberkochen, Germany).

## Pharmacological and genetic inhibition of α-syn aggregation

For the pharmacological inhibition of α-syn aggregation assay, HEK-293T cells were transfected using calcium phosphate with 0.5 μg LIPA-Empty, 0.6 μg LIPA-α-syn, and 1 μg LIPA-α-syn$^{\Delta NAC}$. Twenty-four hours posttransfection, cells were treated with baicalein (20 μM) (465119; Sigma-Aldrich, St. Louis, MO, USA), myricetin (5 μM) (70050, Sigma-Aldrich, St. Louis, MO, USA), naringenin (5 μM) (52186, Sigma-Aldrich, St. Louis, MO, USA), daidzein (20 μM) (D7802; Sigma-Aldrich, St. Louis, MO, USA), or DMSO (D8418; Sigma-Aldrich, St. Louis, MO, USA) (as a negative control), for a period of 12 hours, during which the cells

were subjected to blue light. The percentage of cells exhibiting mCherry aggregates were then assessed at different time intervals (0, 3, 6, 9, 12, 24, 36, and 48 hours) post-blue light illumination. When examining genetic inhibition of LIPA aggregate formation, HEK-293T cells were cotransfected using calcium phosphate transfection with (0.5 μg LIPA-Empty, or 0.6 μg LIPA-α-syn or 1 μg LIPA-α-syn$^{\Delta NAC}$) and the following DNA plasmids (0.6 μg of β-syn, or 0.6 μg of mouse α-syn, or 0.5 μg of GFP or 0.5 μg of empty pCDNA as a negative control). Twenty-four hours posttransfection, cells were exposed to blue light for 12 hours and were then placed in the dark. Percentage of cells exhibiting mCherry aggregates was then assessed at different time intervals (0, 3, 6, 9, 12, 24, 36, and 48 hours) post-blue light illumination. For all experiments, a total of 8 to 10 images/condition were collected using a fluorescent microscope (20X objective) (Nikon Eclipse 80i) (Nikon Instruments, Melville, NY, USA) and a total of 220 to 250 cells were quantified by 2 blinded experimenters using the ImageJ software.

## Filter retardation blotting assay

HEK-293T cells were transiently transfected with the LIPA constructs (LIPA-Empty, LIPA-α-syn, and LIPA-α-syn$^{\Delta NAC}$) as previously described in the Cell culture and DNA transient transfection protocol. After transfection, cells were collected at different time points postillumination with blue light (0.8 mW/mm$^2$), as indicated in the figures. Cells were lysed with a Dremel tissue homogenizer (BioSpec Products, Bartlesville, OK, USA), in lysis buffer (PBS-T 0.05% + Protease inhibitor, Phosphatase inhibitor II, and Phosphatase inhibitor III (P8340, P5726, P0044; Sigma-Aldrich, St. Louis, MO, USA) and phenylmethylsulfonyl fluoride (PMSF) (PMSF-RO; Sigma-Aldrich, St. Louis, MO, USA,). Cell lysates were centrifuged at 2,000$g$ 4˚C for 10 minutes, and the supernatant was collected and diluted in lysis buffer containing 1% SDS and incubated at RT for 10 minutes. The vacuum manifold (Core LifeScience, Niguel, CA, USA) was prepared by using thin filter paper presoaked in water and placed on the manifold. A cellulose acetate membrane (pore size 0.2 μM) (CA022005; SterliTech, Kent, WA, USA) was soaked in PBS containing 1% SDS and placed on top of the filter paper on the manifold. The manifold was tightly closed, and samples were loaded in triplicates into the wells. The samples were then filtered through the membrane by application of a vacuum. After filtration, the membrane was washed 2 times (5 minutes) with PBS containing 0.1% SDS. For loading control, approximately 30% of total sample volume was aliquoted before the addition of SDS, and the samples were loaded on PBS-soaked nitrocellulose membranes (pore size 0.2 μM) (1620112; Bio-Rad, Hercules, CA, USA), which was then placed on top of the filter paper on the manifold. After filtration, the membrane was washed 2 times (5 minutes) with PBS only. The immunoblotting was carried out as described in the SDS-PAGE protocol. At least 3 independent experiments were analyzed for the filter retardation assays.

## Western blot (SDS-PAGE)

Twenty-four hours posttransfection, HEK-293T cells were washed with PBS and the total protein fraction was extracted by dissolving the cell pellet in 2× Laemmli buffer (0.125 M Tris–HCl (pH 6.8), 20% glycerol, 0.2% 2-mercaptoethanol, 0.004% bromphenol blue, 4% SDS) and incubated at 95˚C for 20 minutes for DNA denaturation. Approximately 5 μl of the total protein fraction (corresponding to 25 to 35 μg of proteins) was loaded per well in a 10% SDS-PAGE gel. Gels were run at 110 V for 75 minutes, and the proteins were transferred to nitrocellulose membranes using the Trans-Blot Turbo Transfer system (Bio-Rad, Hercules, CA, USA). The membranes were dried for 1 hour at RT, then incubated in Odyssey blocking buffer (927–40000; LI-COR, Lincoln, NE, USA) at RT for 1 hour prior to O.N incubation with primary antibodies in the blocking solution (see Table 1). Membranes were then washed 3

times with PBS-Tween 0.1% (PBS-T) (10 minutes), incubated with the appropriate secondary antibodies, either 680RD-conjugated or 800W-conjugated (LI-COR Lincoln, NE, USA) (see Table 1) and finally washed 3 times (10 minutes) with PBS-T. Visualization and quantification were carried out with the LI-COR Odyssey scanner and software (LI-COR Lincoln, NE, USA). At least 3 independent experiments were analyzed for SDS-PAGE experiments.

## Co-immunoprecipitation assay

Whole cell lysates, from HEK-293T cells overexpressing LIPA constructs and exposed to blue light for 12 hours, were extracted in lysis buffer (PBS-T 0.05%+ Protease inhibitor, Phosphatase inhibitor II, Phosphatase inhibitor III, and PMSF, using a Dremel tissue homogenizer (BioSpec Products, Bartlesville, OK, USA). Samples were sequentially centrifuged at 4˚C (500$g$ 5 minutes and 1,000$g$ 5 minutes). The supernatant was collected and 10% of the volume was kept as input sample and diluted in 4× Laemmli lysis buffer. The remaining supernatant was then incubated with 5 µg of mCherry antibody for 30 minutes at 4˚C with rotation. During this incubation time, dynabeads Protein G (10003D; Life Technologies, Carlsbad, CA, USA) were resuspended by tilting the vial several times to ensure proper mixing of the beads with the solution. A volume of 70 µl (per 300 µl of sample) of dynabeads was transferred to clean microcentrifuge tubes, equilibrated 3 times (10 minutes) in 300 µl of PBS-T 0.05% + Protease inhibitors. Dynabeads were resuspended with the antibody–sample mixture and then incubated with rotation at 4˚C for 2 hours. After incubation, tubes were placed on the magnet and the supernatant was discarded. Beads were then washed with the PBS-T 0.05% + Protease inhibitors 3 times (10 minutes) at 4˚C with rotation, removing the supernatant between washes. At the last wash, samples were transferred to a new clean microcentrifuge tube to avoid elution of proteins bound to the tube wall. Dynabeads were then eluted by resuspending beads in 50 µl of 2× Laemmli lysis buffer. Pull-down and input samples were heated at 95˚C for 10 minutes, and samples were then subjected to immunoblotting as described in the western blot protocol. At least 3 independent experiments were analyzed for the co-immunoprecipitation assay.

## Purification of LIPA-α-syn aggregates for seeding assay in conditioned media

LIPA-α-syn aggregates were purified from HEK-293T cells transiently overexpressing LIPA-α-syn plasmid. Briefly, after 24 hours of exposure to blue light to induce the formation of LIPA-α-syn inclusions, cells were washed with PBS, collected by scrapping in lysis buffer (PBS + 1% of phosphatases and proteases inhibitors (B15002 and B14001; Bimake, Houston, TX, USA) and then lysed by sonication (2 × 10 seconds at 10% power) (Model 100 Dismembrator) (Thermo-Fisher Scientific, Waltham, MA, USA). Cell lysates were clarified by sequential centrifugation at 4˚C (500$g$ 5 minutes and 1,000$g$ 5 minutes), the supernatant was collected, and the total protein concentration was determined using Pierce BCA Protein Assay Kit (ThermoFisher Scientific, Waltham, MA, USA, 23225). As a negative control, LIPA-α-syn monomers were extracted from HEK-239T cells overexpressing LIPA-α-syn plasmid and not exposed to blue light.

For the seeding experiment, N2a and HEK-239T cells were plated and transiently transfected with α-syn-GFP plasmid. Twenty-four hours posttransfection, cell lysates containing LIPA-α-syn aggregates or monomers were sonicated on ice (60 pulses at 10%), then 50 µg of total protein were transduced to the media using Lipofectamine 2000 protocol. As a positive control, 70 nM of preformed α-syn recombinant fibrils was transduced to the media. Twenty-four hours posttransduction of the purified proteins, cells were washed with PBS and incubated for 3 days in fresh media, fixed and imaged.

For the seeding of LIPA-α-syn by PFF-488, HEK-293T cells were transfected using calcium phosphate with 0.5 μg LIPA-Empty, 0.6 μg LIPA-α-syn, or 1 μg LIPA-α-synΔNAC. Cells were simultaneously incubated with 140 nM PFF-488. Fifty hours postincubation, the cells were fixed and processed for imaging. For the seeding of PFF-488 by LIPA-α-syn, HEK-293T cells were transfected using calcium phosphate with 0.5 μg LIPA-Empty, 0.6 μg LIPA-α-syn, or 1 μg LIPA-α-syn$^{\Delta NAC}$. Cells were concurrently incubated with 140 nM PFF-488. Twenty-four hours posttransfection, cells were exposed to 12 hours of light. After illumination, cells were fixed and processed for imaging.

Pffs were provide by Dr. Edward A. Fon's and Dr. Thomas M. Durcan's teams and were produced, purified, and characterized according to protocols optimized at the Montreal neurological Institute and Hospital, Early drug Discovery platform (https://zenodo.org/record/3738335#.Xv94EC2z2up).

## Purification of LIPA α-syn aggregates for in vitro seeding experiments

HEK-293T cells were transiently transfected with the LIPA-α-syn construct using the calcium phosphate transfection method as described in the Cell culture and DNA transient transfection section. Twenty-four hours posttransfection, cells were exposed or not to blue light for a period of 24 hours. After blue light exposure, cells were washed once with PBS, and 500 μL of lysis buffer containing (PBS, 0.05% Triton X-100 + 1% of phosphatases and proteases inhibitors was used. The cell lysates were subjected to sonication (40% intensity for 20 seconds) using (Fisherbrand Model 505 sonicator) (ThermoFisher Scientific, Waltham, MA, USA).

The cell lysates were then clarified by sequential centrifugation at 4°C (500$g$ 5 minutes and 1,000$g$ 5 minutes), giving a clarified lysate. The clarified lysate was then ultracentrifuged at 150,000$g$ for 1 hour at 4°C, and the pellet was resuspended in 150 μl of PBS. The pellet was then subjected to sonication (30% intensity, 1 second on and 1 second off, for 45 seconds) using (Fisherbrand Model 505 sonicator). When the pellet was too big, this sonication step was performed twice.

The purified LIPA-α-syn aggregates were characterized by western blot and filter retardation assay using α-syn specific antibodies. Protein concentration was determined using Pierce BCA Protein Assay Kit (23225; ThermoFisher Scientific, Waltham, MA, USA); additionally, western blots were performed to validate protein amounts of LIPA α-syn aggregates, compared to purified recombinant monomeric α-syn. Recombinant monomeric α-syn were provide by Dr. Edward A. Fon's and Dr. Thomas M. Ducran's teams and were produced, purified, and characterized according to protocols optimized at the Montreal neurological Institute and Hospital, Early drug Discovery platform (https://zenodo.org/record/3738335#.Xv94EC2z2up).

## PMCA (protein misfolding cyclic amplification)

PMCA experiments were performed according to previously described protocols [34,61]. Briefly, samples of LIPA α-syn aggregates, and LIPA-α-syn monomers (negative control) were prepared and purified as described in the purification of LIPA α-syn aggregates section. PMCA was performed using 1 μg/μl of recombinant monomeric α-syn as substrates, and 1% of LIPA α-syn aggregates and monomers were added as exogenous seeds. Experimental controls included recombinant monomeric α-syn as substrates and purified α-syn PFFs added to a final concentration of 1%. Samples were prepared in the conversion buffer (1% Triton X-100 and 150 mM NaCl in PBS). Samples were prepared to a final volume of 200 μl and were then subjected to cycles of 20-second sonication (Amplitude 50%) and 29-minute 40-second incubation at 37°C for up to 32 cycles. Aliquots taken at different time points were measured for Th-T assay, in addition to a filter retardation blotting assay. Th-T fluorescence reading was

carried out with a Th-T concentration of 10 μM, and 2.5 μM of PMCA product was added to 50 μl of a pH 8.5 buffer containing 50 mM glycine. A plate reader (BioTek Cytation 5 Multi-Mode Readers) (BioTek, Winooski, VT, USA) was used to measure Th-T fluorescence at an excitation wavelength of 450 nm and an emission wavelength of 485 nm. Filter retardation assays were also performed using (30 μl) aliquots and aggregate formation was validated using α-syn specific antibodies. The Th-T assays from PMCA experiments were performed in triplicates.

## RT-QuIC (real-time quaking-induced conversion)

RT-QuIC experiments were performed according to previously described protocols [62]. Briefly, samples of LIPA α-syn aggregates, and LIPA-α-syn monomers (negative control) were prepared and purified as described in the purification of LIPA α-syn aggregates section. Experimental controls included 20 μg/ml of recombinant monomeric α-syn as substrates, and 1.5 μg/ml of purified α-syn Pffs as seeds. Likewise, as seeds, LIPA-α-syn monomers and aggregates were used at concentrations between 1 and 2 μg/ml. A volume of 100 μl of reaction mixtures were pipetted in triplicates in black clear bottom 96-well plates. The reaction mixture was composed of the following: 150 mM NaCl, 1 mM EDTA, 10 μM Th-T, 70 mM SDS, and 20 μg/ml of recombinant monomeric α-syn, in PBS (pH 7.1). Plates were covered with sealing tape and incubated in a plate reader (41˚C, with orbital shaking at 425 rpm during 1 minute followed by a 2-minute rest period). This program was left to run for up to 1 week. The Th-T fluorescence was measured (excitation 435 nm; emission 485 nm) every 30 minutes using (BioTek Cytation 5 Multi-Mode Readers) plate reader. For RT-QuIC, experiments were performed at least 3 times with 3 replicates per experiment.

## Transmission electron microscopy

HEK-293T cells overexpressing LIPA-α-syn and exposed to blue light for 12 hours were fixed in 0.5 mL of 3.5% acrolein and 4% PFA in 0.05 M phosphate-buffered saline (PBS; pH 7.4), overnight at 4˚C. Cells were then washed 3 times in PBS to remove excess fixative. Cell pellets were mixed gently with 125 μL of 4% agarose, kept at 4˚C until solid, and then cut into 50 μm sections using a Leica VT1000S vibratome (Leica Biosystems, Concord, ON, Canada). Sections were washed 3 times in PBS for 10 minutes and were incubated in 3% potassium ferrocyanide and 2% aqueous osmium tetroxide in 0.1 M phosphate buffer (pH 7.4) for 1 hour at RT. After 5 washes (3 minutes) with ddH$_2$O, sections were incubated 20 minutes in a fresh solution of thiocarbohydrazide (1% w/v) at RT, then washed again 5 times (3 minutes) with ddH$_2$O, incubated 30 minutes in 2% aqueous osmium tetroxide, and washed 5 times (3 minutes) with ddH$_2$O. Sections were dehydrated using sequential alcohol washes followed by propylene oxide and embedded in Durcupan resin, infiltrated between ACLAR sheets overnight at RT, then polymerized in the oven at 55˚C for 3 days. Ultrathin sections were generated at approximately 65 nm using a Leica UC7 ultramicrotome. Images of 12 to 13 cells per experimental condition were randomly acquired at 9,300X using a FEI Tecnai Spirit G2 transmission electron microscope (ThermoFisher Scientific, Waltham, MA, USA) operating at 80 kV and equipped with a Hamamatsu ORCA-HR digital camera (10 MP).

α-Syn aggregates, purified from HEK-293T cells according to the protocol described in the section (Purification of LIPA α-syn aggregates for in vitro seeding experiments), were characterized using a negative staining protocol and analyzed using an electron microscope. Purified α-syn aggregates were pipetted on a 200 mesh copper carbon grid (3520C-FA, SPI Supplies) and fixed with 4% PFA for 1 minute, followed by staining with 2% acetate uranyl (22400–2, EMS) for 1 minute. α-Syn aggregates were visualized using a transmission electron microscope

(FEI Tecnai G2 Spirit Twin 120 kV TEM) coupled to a camera (AMT XR80C CCD Camera or Gatan Ultrascan 4000 4k × 4k CCD Camera model 895, respectively). Unilamellar-like vesicles were considered as a liposome containing a single phospholipid bilayer sphere and were classified in small unilamellar-like vesicles (SUVs, diameter <100 nm) and large unilamellar-like vesicles (LUVs, diameter >100 nm). Autophagic-vacuolar-like structures were identified as a vesicle with a membrane enclosed. All TEM experiments were performed in triplicates.

## STED microscopy

STED images were acquired with a Leica TCS SP8 STED 3X microscope (equipped with white-light laser, 405-nm diode laser and HyD detectors) (Leica Biosystems, Concord, ON, Canada) by sequential scanning between stacks with a 592-nm (Alexa Fluor 488) or 660-nm (mCherry) depletion using the following settings: objective, HC PL/APO CS2 ×100/1.40 oil STED White; immersion oil, Leica Type F Immersion liquid (refractive index, 1.5180); zoom factor, 6.00; scan speed, 600 Hz; line average, 8 to 16 (Alexa Fluor 488 or mCherry, respectively); time gate, 0 to 3.5 ns; STED 3D, 50%; depletion laser intensity, 5% to 10% (Alexa Fluor 488, depending on the signal) or 30% (mCherry). Laser power and gain were set to optimize signal-to-noise ratio and avoid saturation using the QLUT Glow mode. Sizes of pixel, pinhole, and z-step were set to optimize resolution or to oversample in the case of images to be deconvolved. Deconvolution was performed using Huygens Professional (Scientific Volume Imaging, Hilversum, the Netherlands) using a theoretical point spread function, manual settings for background intensity, and default signal-to-noise ratio. Color balance, contrast, and brightness were adjusted with ImageJ.

## Stereotaxic injections of AAV viral particles, implantation of wireless optogenetic devices, and optogenetic stimulation

Three-month-old C57/BL6 mice were obtained from Charles River Laboratories and habituated for a 7-day period before any handling. Mice were housed with on 12 hours light/dark cycle with ad libitum access to food and water. All animal experiments were approved by the Animal Welfare Committee of Université Laval in accordance with the Canadian Council on Animal Care policy, protocol 2020–641 to AO. All experiments for mini-endoscopic imaging in combination with optogenetic stimulation were approved by the Animal Welfare Committee of Université Laval, protocol 2019–020 to AS.

Mice were anesthetized with 2% isoflurane-$O_2$ and were placed in the stereotaxic frame (David Kopf Instruments, Los Angeles, CA, USA). The top of the skull was incised with a scalpel, and the tissues were cleared to visualize the interaural point and the Bregma. After piercing the skull with drill, mice received a unilateral injection of 2 μL of viral suspension at the rate of 0.2 μL/min, which corresponds to a total viral load of $2 \times 10^{10}$ GC, using automatic pumps (David Kopf Instruments). Injections were performed in the SNc and in the neostriatum using the following coordinates, respectively: (−3.08 mm posterior, −1.5 mm lateral, −4.25 mm ventral) and (+0.2 mm posterior, −2 mm lateral, −3 mm ventral). Injections were made using a 10-μL syringe (Hamilton, Reno, NV, USA) and 30-gauge needle. A needle was placed in the injection site 3 minutes preinjection and left for an additional 5 minutes postinjection before it was slowly withdrawn. During the same surgery session, mice were implanted with the wireless optogenetic devices (Eicom/Amuza, San Diego, CA, USA) in the SNc or the neostriatum, using the following coordinates, respectively: (−3.08 mm posterior, −1.5 mm lateral, −4.20 mm ventral) and (+0.2 mm posterior, −2 mm lateral, −3 mm ventral).

Fifteen days postsurgery, in vivo light stimulation sessions were performed every second day using the wireless optogenetic devices (Eicom/Amuza). Right before each session, a battery

(Eicom/Amuza) was directly connected to the implant. Stimulation was performed for 1 hour using a pulse generator (Eicom/Amuza) at the power of 1.76 mW/mm$^2$ (measured at the tip of the optical fiber) and a pulse of 10 ms at 20 Hz.

## Behavioral tests

All behavioral tests were completed during the light phase of the light–dark cycle between 8 AM and 4 PM. Mice were habituated to the experimenter and the testing room for several days prior to the start of testing. All group assignments were randomized, and the experimenter was blind to viral vector treatment during testing and blind to both vector group and illumination group during scoring and all group assignments were randomized.

**The cylinder test.** The cylinder test was performed to evaluate the motor impairment induced after a dopaminergic neuronal loss, by quantifying the deficits in using the contralateral forelimb (akinesia) [63–65]. Briefly, mice were placed in a transparent Plexiglas cylinder (15 cm diameter, 12 cm high) surrounded by a mirror to monitor the mouse from all angles and videotaped using a camera (Microsoft LifeCam Cinema) (H5D-00018; Microsoft, Redmond, WA, USA). A total number of 30 forepaw contacts made on the cylinder wall by the ipsilateral or the contralateral (impaired) forelimbs were scored, and the results were expressed as the ratio of contralateral contacts relative to the total contacts made by both forelimbs.

**The Grip Strength test.** Grip Strength (CHATILLON DFE Series) (Ametek, Berwyn, PA, USA) was used to measure the muscle strength of forelimbs as previously described [66]. During testing, the mouse is placed horizontally on the grid to allow gripping of the grid with the forepaws while being supported by the tail. Once the grid is gripped, the mouse is pulled back until the grip on the grid is released and the value on the apparatus is recorded. Mice underwent 3 trials per testing session, and analysis was performed on the average for the 3 trials.

**The rotarod test.** The rotarod test was used to measure motor coordination, endurance, and balance [67]. All mice were pretrained on the rotarod (LE 8200; Panlab Harvard apparatus, Holliston, MA, USA) at baseline to reach a stable performance. At testing, mice were placed on the rotarod for 3 consecutive 3-minute trials, at fixed speed of 8 RPM with 1 minute of rest between each trial. The latency to fall was recorded for each trial and the mean value from each speed and was used for analysis.

**The gait test.** To test for gait abnormalities, footprint gait analysis was performed as previously described [68] with some modifications. Briefly, the hind- and forefeet of the mice were painted with blue (right paws) and orange (left paws) nontoxic paint (Liquid tempera) (SCHOLA, Marieville, QC, Canada) immediately prior to placement in a 45-cm long and 15-cm wide runway coated in newsprint paper. Stride length was measured as the distance of forward movement between paw prints. The mean value of each set of 3 values measuring stride length was used in the analysis.

## Tissue processing, immunohistochemistry, and immunocytochemistry

**Tissue processing—Microtome sections.** Animals were killed with ketamine/xylazine overdose (90 mg/ml/10 mg/ml, injected 0.1 ml/30 g mice), and brains were removed after transcardial perfusion with PBS, followed by 4% PFA-PBS. Brains were postfixed for 4 hours in 4% PFA, then incubated in 25% sucrose-PBS for 24 hours. Postfixation, brains were cut in coronal sections (30 μm thick) with a microtome (SM2000R; Leica, Wetzlar, Germany), and slices were stored at −20˚C in cryoprotection medium.

**Tissue processing—Vibrotome sections.** Animals were killed with ketamine/xylazine overdose (100 mg/ml, injected 0.1 ml/10 g mice), and brains were removed after transcardial

perfusion with 0.9% NaCl, followed by 4% PFA-PBS. Brains were postfixed O.N in 4% PFA. Postfixation, brains were placed in 4% agar, and cut in sagittal sections (100 μm thick), with a vibratome (VT1000S; Leica, Wetzlar, Germany) and stored at 4˚C.

For immunohistochemistry, slices were washed 3 times (10 minutes) with PBS and then incubated for 1 hour at RT in blocking buffer (3% bovine serum albumin (BSA), 0.1% Triton X100-PBS) and then incubated overnight at 4˚C with primary antibodies prepared in blocking buffer (see Table 1). For enzymatic revelation, slices were incubated with biotinylated secondary antibodies (see Table 1). Slices were washed 3 times (10 minutes) with PBS; the VECTASTAIN Elite ABC Kit (PK6100; Vectastain, Burlingame, CA) was then used for enzymatic staining. Slices were incubated in 3,3′-diaminobenzidine tetrahydrochloride (DAB) (D5905; Sigma-Aldrich, St. Louis, MO, USA) and subsequently counterstained with Cresyl Violet acetate stain (Nissl) (C5042; Sigma-Aldrich, St. Louis, MO, USA). For microscopy imaging, slices were mounted on Superfrost Plus microscope slides (12-550-15; Fisherbrand, Waltham, MA, USA), dried and finally mounted in DTX Mounting Medium (13512; EMS, Hatfield, PA, USA).

For immunofluorescence, slices were washed 3 times (10 minutes) with PBS and then incubated for 1 hour at RT in blocking buffer (3% BSA, 0.1% Triton X100-PBS) and then incubated overnight at 4˚C with primary antibodies prepared in blocking buffer (see Table 1). The slices were then washed 3 times (10 minutes) with PBS and then incubated, for 2 hours at RT, with the appropriate secondary antibodies in 0.1% Triton X100-PBS conjugated to Alexa Fluor-488 or Alexa Fluor-633 (see Table 1). After secondary antibody incubation, slices were washed 3 times (10 minutes) with 0.1% Triton X100- PBS, then incubated in DAPI (1:5,000 for 7 minutes) and finally washed 2 times (10 minutes) with PBS. For nuclear staining, slices were counterstained with DAPI (1:5,000). For microscopy imaging, slices were mounted on Superfrost Plus microscope slides in Fluoromount-G T (17984–25; EMS, Hatfield, PA, USA) mounting media and were left to dry in the dark for 2 days.

TH quantification was performed by evaluating [1] the number of double positive cells over the total number of TH-positive cells, and [2] the number of TH-positive cells with mCherry inclusions over the total number of TH-positive cells. For the quantification of the subtype of cells transduced by AAV-LIPA-α-syn. The same procedure as described above was performed for anti-NeuN, anti-pS129, and anti-mCherry antibodies (see Table 1). The quantification was performed by evaluating the number of double positive cells over the total number of mCherry-positive cells. Images were taken using a Zeiss LSM800 confocal microscope (Zeiss, Oberkochen, Germany) and analyzed using ImageJ software. All quantifiacation was done on slices from 5 to 6 animals per condition.

## Unbiased stereological estimation of dopaminergic neurons in the SNc

Dopaminergic neuronal number was estimated using unbiased stereology according to the optical fractionator principle described by West and colleagues [69]. Briefly, the number of TH-immunoreactive neurons and Nissl-positive cells were determined by evaluating every fourth coronal section (1/4) covering the entire SNc structure. The SNc was delineated at low magnification (20X) and then the dopaminergic neurons were counted under an oil immersion objective (60X). TH+ neurons were counted in a blinded fashion, and the results are expressed as the mean ± standard error of the total number of TH+ neuron in the injected side. Analysis was performed using the MBF Stereo Investigator software (MBF Bioscience, Williston, VT, USA). The parameters used for the stereological analysis were as follows: grid size, 150 × 150 μm; counting frame, 75 × 75 μm; and 2 μm guard zones. Tissue thickness was determined at each counting field. The coefficient of error was <0.1.

## Optogenetic stimulation of the SNc and mini-endoscopic imaging of the dorsal striatum

C57Bl/6NCrl mice (2-month-old) (Charles Rivers, Wilmington, MA, USA) received a stereotaxic injection of AAV2/6-LIPA-α-syn-mCherry in the SNc and AAV2/5 CAG-GCaMP6s injection in the striatum. Stereotaxic injections were performed using the following coordinates (with regards to bregma): AP −3.5, ML +/−1.2, DV −4 for SNc, and AP −3.5, ML 1.2, DV −4 and AP +1.5, ML +/−1.2, DV −2.5 for dorsal striatum. Approximately 2 to 3 weeks later, animal received a second surgery during which a double optogenetic/mini-endoscopic cannula was implanted in the way that optic fibers to be positioned just above SNc and GRIN lens just above the dorsal striatum. The mice were anesthetized with 2% to 3% isoflurane, and their body temperature was maintained at 37.5˚C using an infrared heating blanket (Kent Scientific, Torrington, CT, USA). After removing the skin, the skull above SNc and striatum (where virus injections were previously performed) was drilled and a 250-μm optic fiber as well as a 500-μm GRIN lens were slowly implanted (0.3 mm every 30 seconds/1 minute) into the SNc and dorsal striatum, respectively. The implant was custom-designed with the following characteristics: GRIN lens protrusion length 2.9 mm, fiber protrusion length 4.0 mm, and pitch between the 2 implants 5.0 mm. The exposed optic fiber was protected with Kwik-Seal (World Precision Instruments, Sarasota County, FL, USA) and Metabond cement (Parkell, Brentwood, NY, USA) was used to fix the implant on top of the animal's head. Following 3 weeks of recovery, $Ca^{2+}$ recordings were performed twice for 10 minutes in the striatum in freely behaving animals every second day. Excitation was performed using a 465-nm LED light source, at a frame rate of 10 fps and an exposure of 100 ms. Starting on day 4 following $Ca^{2+}$ imaging, animals also received a 1-hour optogenetic stimulation in the SNc prior to $Ca^{2+}$ recordings, following the same pattern of stimulation as previously described in the Stereotaxic injections of AAV viral particles, implantation of wireless optogenetic devices, and optogenetic stimulation section. At the end of the 7-day stimulation protocol, animals were imaged every second day for up to 10 days after the end of the stimulation.

## $Ca^{2+}$ dynamics analysis

Acquired images were first time-averaged in 200 ms bins using the ImageJ software. Using the Doric Neuroscience Studio software (Doric Lenses, Quebec, QC, Canada), images were corrected in XY for brain motion, a ΔF/F0 processing was first applied, and the background was subtracted. ROIs were traced around cell bodies, and mean fluorescence intensity values were extracted at every time point. $Ca^{2+}$ responses were analyzed using a custom-written script in MATLAB (The MathWorks, Natick, MA, USA). For each ROI, $Ca^{2+}$ events with an amplitude higher than a threshold of 2.5 times the standard deviation ($2.5 \times SD$) of the mean fluorescence of the trace were automatically detected. Values over this threshold were then removed from the traces, and the $2.5 \times SD$ was applied for a second and third time, allowing for the detection of peaks of smaller amplitude. For the 3 levels of peak detection, temporal filtering was then applied, with a 2-second "refractory period" set between 2 consecutive peaks. Mean amplitude and frequency of $Ca^{2+}$ events detected this way were calculated, as well as the percentage of synchronized cells across time (i.e., cells exhibiting a $Ca^{2+}$ event at the same frame).

## LDH cytotoxicity assay

Total LDH activity of HEK-293T cells expressing LIPA constructs exposed to 24 hours of continuous light stimulation was assessed using the Pierce LDH Cytotoxicity Assay Kit (88953; ThermoFisher Scientific, Waltham, MA, USA). Approximately 10,000 cells were plated in

triplicates in 96-well plates and incubated overnight at 37˚C, 5% $CO_2$; cells were then exposed to blue light for a period of 24 hours, and LDH activity was measured as described by the manufacturer. Briefly, after light exposure, the supernatant was harvested from both light and non-light exposed cells, by transferring 50 μL of each sample medium to a 96-well flat bottom dark plate in triplicate wells. A volume of 50 μL of Reaction Mixture was then added to each sample well and mixed by gentle tapping. The plate was then incubated at RT for 30 minutes protected from light. Absorbance was then measured at 490 nm and 680 nm using the Synergy HT plate reader (BioTek, Winooski, VT, USA). The 680-nm absorbance value (background signal from instrument) was subtracted from the 490-nm absorbance. To determine LDH activity or leakage presented as A.U, we used the following formula: Leakage = (LDH leakage in light exposed cells or non-light exposed cells − Spontaneous LDH activity) / (Maximum LDH activity − Spontaneous LDH activity). The LDH positive control provided from the manufacturer was used as our positive control. At least 3 independent experiments were analyzed for the LDH activity assay.

## Statistical analysis

All cell-based assays were performed in at least 3 independent experiments. Statistical analysis was performed using one-way ANOVA followed by Tukey's multiple comparisons test.

For analysis of behavioral data, statistics were performed using two-way ANOVA followed by Tukey's post hoc tests. Across time, data were analyzed with a linear mixed-effects model of viral vector and implant. Within-subjects variance was controlled for by including random effects of intercept and slope for each mouse. The model was estimated using maximum likelihood and contrast comparisons were performed to determine the effect of viral vector and implants at each time point. Analyses were performed using RStudio version 3.4.1 with nlme version 3.1–131.

For the analysis of $Ca^{2+}$ dynamic in the striatal neurons, we used two-tailed paired Student $t$ test to compare the results between days of imaging.

All values were expressed as the means ± SEM, and the software used for the statistical analysis was Prism v.6 (GraphPad, La Jolla, CA, USA). $p < 0.05$ was required for rejection of the null hypothesis.

## Supporting information

**S1 Fig. Blue light stimulation does not induce cell toxicity in cell culture and is required for the induction of LIPA aggregates.** (**A**) Western blot and quantification of the protein levels (mCherry) of the different LIPA constructs showing similar expression levels after DNA normalization during the transient transfection ($n = 3$). The data are presented as the means ± SEM. (**B**) Cell toxicity assay assessed by quantifying the extracellular release of cytosolic LDH from HEK-293T cells overexpressing LIPA constructs exposed (blue histograms) or not (gray histograms) to 24 hours of continuous blue light stimulation at 0.8 mW/mm$^2$ ($n = 4$). The data are presented as the means ± SEM. (**C**) In the absence of blue light stimulation, LIPA constructs exhibit diffuse cytosolic expression without detectable protein aggregates ($n = 5$) (scale bar = 10 μm). (**D**) HEK-293T cells overexpressing α-syn-mCherry, exposed or not to the blue light, did not display inclusion formation ($n = 3$) (scale bar = 10 μm). The underlying data for (**A**) and (**B**) can be found in S1 Data. LDH, lactate dehydrogenase; LIPA, light-inducible protein aggregation.
(TIF)

**S2 Fig. CIB1-GFP overexpression rescues LIPA-α-syn$^{\Delta NAC}$ light-induced aggregation.** (**A**) Confocal microscopy images illustrating HEK-293T cells overexpressing LIPA-α-syn$^{\Delta NAC}$ alone or with CIB1-GFP exposed to blue light for 2 hours. Only cells overexpressing LIPA-α-syn$^{\Delta NAC}$ and CIB1 showed mCherry-positive inclusions. These inclusions were also positive for GFP, revealing the coaggregation of the 2 proteins ($n = 4$) (scale bar = 5 μm). (**B**) Quantification of mCherry-positive LIPA inclusions, showing a significant increase in the proportion of cells with LIPA aggregates under the condition of both LIPA-α-syn$^{\Delta NAC}$ and CIB1-GFP ($n = 4$). The data are presented as the means ± SEM. **** $p \leq 0.0001$, LIPA-α-syn$^{\Delta NAC}$ + CIB1-GFP + light vs LIPA-α-syn$^{\Delta NAC}$ alone + light and $^{\$\$\$\$} p \leq 0.0001$, LIPA-α-syn$^{\Delta NAC}$ + CIBN1-GFP + light vs LIPA-α-syn$^{\Delta NAC}$ + CIB1-GFP–light. The underlying data for (**B**) can be found in S1 Data. α-syn, α-synuclein; LIPA, light-inducible protein aggregation. (TIF)

**S3 Fig. Nonilluminated cells overexpressing LIPA-α-syn do not exhibit LB-like inclusions.** (**A**) Confocal microscopy images of representative HEK-293T cells and (**B**) hiPSC-derived neurons overexpressing LIPA-α-syn not exposed to blue light stimulation. Staining with antibodies against α-syn (BDlab), pS129, thioflavin S, ubiquitin, HSP70, and p62 revealed the absence of LB-like inclusions ($n = 5$) (scale bars = 10 μm and 5 μm in **A** and **B**, respectively). α-syn, α-synuclein; hiPSC, human-induced pluripotent stem cell; LB, Lewy bodies; LIPA, light-inducible protein aggregation; pS129, phosphorylated α-syn at S129. (TIF)

**S4 Fig. Absence of authentic LBs markers in HEK-293T cells overexpressing the LIPA-Empty or LIPA-α-syn$^{\Delta NAC}$ constructs.** (**A**) Confocal microscopy images of representative HEK-293T cells overexpressing the LIPA-empty construct (+/−light) and (**B**) LIPA-α-syn$^{\Delta NAC}$ construct (+/−light). Staining with antibodies against α-syn, pS129, Thio. S, Ub, HSP70, and p62 revealed the absence of LB-like inclusions ($n = 3$) (scale bars = 5 μm). α-syn, α-synuclein; LB, Lewy bodies; LIPA, light-inducible protein aggregation; pS129, phosphorylated α-syn at S129; Thio. S, thioflavin S; Ub, ubiquitin. (TIF)

**S5 Fig. Absence of authentic LBs markers in hiPSC-derived neurons overexpressing LIPA-Empty or LIPA-α-syn$^{\Delta NAC}$ constructs.** (**A**) Confocal microscopy images of representative hiPSC-derived neurons overexpressing LIPA-empty construct (+/−light) and (**B**) LIPA-α-syn$^{\Delta NAC}$ construct (+/−light). Staining with antibodies against α-syn, pS129, Thio. S, Ub, HSP70, and p62 revealed the absence of LB-like inclusions ($n = 3$) (scale bars = 5 μm). α-syn, α-synuclein; hiPSC, human-induced pluripotent stem cell; LB, Lewy bodies; LIPA, light-inducible protein aggregation; pS129, phosphorylated α-syn at S129; Thio. S, thioflavin S; Ub, ubiquitin. (TIF)

**S6 Fig. Representative STED images illustrating the presence of vesicular markers within LIPA-α-syn inclusions only.** (**A**) HEK-293T cells overexpressing LIPA-α-syn were exposed to blue light for 12 hours and stained with antibodies against markers of endogenous vesicles. (**B**) HEK-293T cells overexpressing LIPA-Empty were exposed to blue light for 12 hours and stained with antibodies against markers of endogenous vesicles. (**C**) HEK-293T cells overexpressing LIPA-α-syn$^{\Delta NAC}$ were exposed to blue light for 12 hours and stained with antibodies against markers of endogenous vesicles. Staining for endogenous vesicles included EEA1, marker of early endosomes; LAMP1 and LAMP2, markers of lysosomes/late endosomes; and staining for CD9, marker of early exosomes ($n = 3$) (scale bar = 10 μm). α-syn, α-synuclein;

LIPA, light-inducible protein aggregation; STED, stimulated emission depletion.
(TIF)

**S7 Fig. Stability and seeding capacity of LIPA-α-syn aggregates.** (**A**) Schematic representation of the experimental paradigm to test LIPA inclusion stability. (**B**) Time course of LIPA-empty aggregate dissociation in the presence of the α-syn aggregation small-molecule inhibitors baicalein and myricetin ($n = 3$). (**C**) Time course of LIPA-Empty aggregate dissociation after the overexpression of β-syn and mα-syn ($n = 3$). (**D**) Time course of LIPA-α-syn aggregate dissociation in the presence of small molecules not affecting α-syn aggregation, naringenin or daidzein ($n = 3$). (**E**) Time course of LIPA-α-syn aggregate dissociation after the overexpression of GFP ($n = 3$). The data are presented as the means ± SEM. (**F**) Three-dimensional reconstitution illustrating the seeding of α-syn-GFP by LIPA-α-syn aggregates as observed after 12 hours of blue light stimulation (scale bars = 0.5 μm). (**G**) Representative confocal images of HEK-293T cells overexpressing α-syn-GFP alone or the LIPA-Empty or LIPA-α-syn$^{\Delta NAC}$ constructs (scale bar = 5 μm). (**H** and **I**) Seeding capacity of LIPA-α-syn aggregates of α-syn-GFP when added to the culture medium with N2a or HEK-293T cells ($n = 3$), similar observation were collected when α-syn-Pffs were added to the culture medium (scale bars = 10 μm). (**J**) Representative confocal images of HEK-293T cells overexpressing LIPA constructs and cultured in the presence of α-syn Pff-488 added to the culture medium. Results showed that α-syn Pff-488 were able to seed the aggregation of LIPA-α-syn aggregates in the absence of light stimulation. In contrast, no seeding effect was observed in cells expressing the LIPA-α-syn$^{\Delta NAC}$ or LIPA-Empty constructs ($n = 3$) (scale bars = 10 μm). (**K**) RT-QuIC analysis illustrating the kinetics of recombinant α-syn aggregation in the presence of purified LIPA-α-syn$^{\Delta NAC}$ (+/−light), purified LIPA-Empty (+/−light), and recombinant α-syn Pffs. The average ThT fluorescence intensity was plotted against time ($n = 3$). The data are presented as the means ± SEM. The underlying data for (**B**), (**C**), (**D**), (**E**), and (**K**) can be found in S1 Data. α-syn, α-synuclein; β-syn, β-synuclein; LIPA, light-inducible protein aggregation; mα-syn, mouse α-syn; Pff, preformed fibril; RT-QuIC, real-time quaking-induced conversion; ThT, thioflavin T.
(TIF)

**S8 Fig. Blue light stimulation does not induce neuronal toxicity per se in vivo and is required for the induction of LIPA-α-syn aggregation in mouse midbrains.** (**A**) Microscopy images illustrating the density of TH+ dopaminergic neurons in WT mice implanted with optogenetic devices in the midbrain and exposed to blue light for 30 minutes/day or 1 hour/day every other day for 7 days (scale bar = 1 mm). (**B**) Stereological unbiased quantification of dopaminergic neurons after exposure to blue light for 30 minutes/day or 1 hour/day every other day for 7 days, showing the absence of neuronal loss after light stimulation ($n = 4$–6 mice). The data are presented as the means ± SEM. (**C**) Representative confocal images of midbrain neurons (scale bar = 200 μm) and (**D**) histograms showing the proportion of neurons (NeuN+) overexpressing the LIPA constructs (mCherry+) ($n = 5$–6 mice). (**E**) Confocal microscopy images of representative midbrain dopaminergic neurons (scale bar = 200 μm) and (**F**) histograms showing the proportion of the dopaminergic neurons (TH+) overexpressing the LIPA constructs (mCherry+) ($n = 5$–6 mice). (**G**) Histograms showing the proportion of dopaminergic neurons (TH+) depicting LIPA-α-syn inclusions (mCherry+) in the presence or absence of light stimulation ($n = 5$–6 mice). (**H**) Confocal microscopy images of representative midbrain neurons not exposed to blue light stimulation showing the absence of LIPA-α-syn aggregates ($n = 4$ mice) (scale bar = 5 μm). The underlying data for (**B**), (**D**), (**F**), and (**G**) can be found in S1 Data. α-syn, α-synuclein; LIPA, light-inducible protein aggregation; SN,

substantia nigra; WT, wild-type.
(TIF)

**S9 Fig. LIPA-α-syn-induced inclusions recapitulate authentic LB features in the striatum.**
(**A**) Experimental design of the overexpression and induction of LIPA-α-syn aggregation in
the striatum of WT mice. (**B**) Representative confocal microscopy images of striatal neurons
with LIPA-α-syn aggregates exhibiting authentic LB markers: α-syn (BDlab and FL140 anti-
bodies), α-syn pS129, thioflavin S, ubiquitin, and p62 ($n$ = 5 mice) (scale bar = 5 μm). (**C**) Con-
focal microscopy images of representative striatal neurons overexpressing LIPA-α-syn not
exposed to blue light stimulation and stained with authentic LB markers: α-syn (BDlab),
pS129, thioflavin S, ubiquitin, HSP70, and p62 ($n$ = 5 mice) (scale bar = 5 μm). AAV, adeno-
associated virus; α-syn, α-synuclein; LB, Lewy bodies; LIPA, light-inducible protein aggrega-
tion; pS129, phosphorylated α-syn at S129; WT, wild-type.
(TIF)

**S10 Fig. LIPA-α-syn inclusions, but not LIPA-α-syn$^{\Delta NAC}$ overexpression, disrupt nigros-
triatal neuronal transmission.** Representative heat maps of Ca$^{2+}$ signals in one animal overex-
pressing (**A**) LIPA-α-syn exposed to blue light, (**B**) LIPA-α-syn without light stimulation, and
(**C**) LIPA-α-syn$^{\Delta NAC}$ exposed to blue light. Analysis was performed before (Baseline), during
(Stimulation day 5), and after (Poststimulation day 10) optogenetic stimulation. The underly-
ing data for (**A**), (**B**), and (**C**) can be found in S1 Data. α-syn, α-synuclein; LIPA, light-induc-
ible protein aggregation.
(TIF)

**S1 Video. Formation of LIPA-α-syn aggregates under blue light stimulation.** HEK-293T
cells overexpressing the LIPA-α-syn construct stimulated with blue light (488 nm, 15.5 μW/
mm$^2$) every 30 sec followed by 5-sec acquisition at 555 nm.
(MOV)

**S2 Video. Formation of LIPA-Empty aggregates under blue light stimulation.** HEK-293T
cells overexpressing the LIPA-Empty construct stimulated with blue light (488 nm, 15.5 μW/
mm$^2$) every 30 sec followed by 5-sec acquisition at 555 nm.
(MOV)

**S3 Video. Absence of LIPA-α-syn$^{\Delta NAC}$ aggregates after blue light stimulation.** HEK-293T
cells overexpressing the LIPA-α-syn$^{\Delta NAC}$ construct stimulated with blue light (488 nm,
15.5 μW/mm$^2$) every 30 sec followed by 5-sec acquisition at 555 nm.
(MOV)

**S4 Video. Subcellular local activation of LIPA-α-syn aggregates.** HEK-293T cell overexpres-
sing the LIPA-α-syn construct stimulated with blue light (488 nm, 15.5 μW/mm$^2$) every 30 sec
followed by 5-sec acquisition at 555 nm, within a 5-μm diameter circle, allowing for the local
formation of LIPA-α-syn aggregates in the cytosol.
(MOV)

**S5 Video. LIPA-α-syn seeds the aggregation of α-syn-GFP.** HEK-293T cells overexpressing
LIPA-α-syn and α-syn-GFP constructs and exposed to blue light (456 nm, 0.8 mW/mm$^2$) for
12 hours and fixed in 4% PFA. Z-stack confocal imaging and animation rendering with Imaris
software showing the seeding effect of LIPA-α-syn and its coaggregation with α-syn-GFP.
(MOV)

**S6 Video. LIPA-α-syn aggregation disrupts nigrostriatal transmission.** Endoscopic live
imaging of striatal neurons infected with AAV-CAG-GCaMP6s in freely behaving mice

before, during, and after induction of LIPA-α-syn aggregates in the midbrain.
(MP4)

**S1 Data. Underlying numerical data for Figs 1C, 1D, 1E, 1F, 1H,1I, 3A, 3B, 3C, 3H, 3I, 3J, 4F, 4G, 4H, 4I, 4J, 4K, 5F, 5G, 5H, S1A, S1B, S2B, S7B, S7C, S7D, S7E, S7K, S8B, S8D, S8F, S8G, S10A, S10B, and S10C.**
(XLSX)

**S1 Raw images. Original scan images for Figs 1D, 1E, 1F, 3E, 3J, and S1A.**
(PDF)

## Acknowledgments

We would like to thank Ms. Frédérique Larroquette from McGill University for her technical assistance.

## Author Contributions

**Conceptualization:** Abid Oueslati.

**Data curation:** Morgan Bérard, Razan Sheta, Sarah Malvaut, Raquel Rodriguez-Aller, Maxime Teixeira, Walid Idi, Marie-Eve Tremblay, Denis Soulet, Francesca Cicchetti, Armen Saghatelyan, Abid Oueslati.

**Formal analysis:** Morgan Bérard, Razan Sheta, Denis Soulet, Armen Saghatelyan, Abid Oueslati.

**Funding acquisition:** Abid Oueslati.

**Investigation:** Abid Oueslati.

**Methodology:** Morgan Bérard, Razan Sheta, Sarah Malvaut, Raquel Rodriguez-Aller, Maxime Teixeira, Walid Idi, Roxanne Turmel, Melanie Alpaugh, Marilyn Dubois, Manel Dahmene, Charleen Salesse, Jérôme Lamontagne-Proulx, Marie-Kim St-Pierre, Esther Del Cid-Pellitero, Marie-Eve Tremblay, Armen Saghatelyan, Abid Oueslati.

**Project administration:** Razan Sheta, Abid Oueslati.

**Resources:** Omid Tavassoly, Wen Luo, Raza Qazi, Jae-Woong Jeong, Thomas M. Durcan, Luc Vallières, Martin Lévesque, Edward A. Fon.

**Supervision:** Abid Oueslati.

**Validation:** Abid Oueslati.

**Writing – original draft:** Morgan Bérard, Razan Sheta, Francesca Cicchetti, Abid Oueslati.

**Writing – review & editing:** Razan Sheta, Abid Oueslati.

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
