## [Editor Report · Decision Letter 0]

29 Sep 2021

Dear Dr Oueslati, 

Thank you for submitting your manuscript entitled "A light-inducible protein aggregation system for modeling proteinopathies and neurodegeneration" for consideration as a Methods and Resources article by PLOS Biology.

Your manuscript has now been evaluated by the PLOS Biology editorial staff as well as by an academic editor with relevant expertise and I am writing to let you know that we would like to send your submission out for external peer review.

**As a note: if your manuscript has been previously reviewed at another journal, PLOS Biology is willing work with those reviews in order to avoid starting from scratch. The decision to submit previous reviews is optional and our ability to use them will depend on the willingness of the previous journal to confirm the content of the reports and share the reviewer identities. Please note that, even if we receive confirmation from the other journal, we might realize that additional/independent reviewers are needed and reserve the right to invite additional referees. However, in our experience, working with previous reviews does save time.

If you would like to send previous reviewer reports to us, please include a point-by-point response to reviewers that details how you plan to address the reviewers’ concerns.

Please re-submit your manuscript within two working days, i.e. by Oct 01 2021 11:59PM. But do let me know if you would like an extension to that timeline, or have any questions. 

Kind regards,

Lucas Smith

Associate Editor

PLOS Biology

lsmith@plos.org

---

## [Decision Letter · Decision Letter 1]

3 Nov 2021

Dear Dr Oueslati,

Thank you for submitting your manuscript "A light-inducible protein aggregation system for modeling proteinopathies and neurodegeneration" for consideration as a Methods and Resources at PLOS Biology. Your manuscript has been evaluated by the PLOS Biology editors, an Academic Editor with relevant expertise, and by several independent reviewers.

The reviews of your manuscript are appended below. The reviewers note that the study is interesting and that it provides a potentially useful tool for the study of Parkinson’s Disease. However, they have also raised a number of important and overlapping concerns which need to be thoroughly addressed for us to consider your manuscript further at PLOS Biology. The reviewers have noted that a number of claims made in the manuscript which should be toned down, and highlight the need for additional controls and more analyses to strengthen and expand the study.

In light of the reviews, we will not be able to accept the current version of the manuscript, but we would welcome re-submission of a much-revised version that takes into account the reviewers' comments. In addressing the reviewer comments, we think that you should pay particularly close attention to the requests for additional controls. We cannot make any decision about publication until we have seen the revised manuscript and your response to the reviewers' comments. Your revised manuscript is also likely to be sent for further evaluation by the reviewers.

We expect to receive your revised manuscript within 3 months. 

**IMPORTANT - SUBMITTING YOUR REVISION**

*Re-submission Checklist*

*Published Peer Review*

*PLOS Data Policy*

*Blot and Gel Data Policy*

Sincerely,

Lucas Smith

Associate Editor

PLOS Biology

lsmith@plos.org

REVIEWS:

Reviewer #1: In this manuscript, the authors established an optogenetic method to produce Lewy body (LB)-like structures. The authors demonstrated that enhancement of interactions between the optogenetic α-synuclein proteins led to cytotoxicity, showing that their model would help address the mechanisms of α-synucleopathy. The authors have carefully characterized their model from in vitro to in vivo, and demonstrated that LIPA-α-syn can produce protein aggregates that recapitulate key disease signatures of LBs. This method provides a nice model of LB formation in Parkinson's disease (PD) and LB-associated proteinopathies.

Major comments:

1: α-synuclein is intrinsically disordered protein (PMID: 32514159) and CRY2-based optogenetic modulations of intrinsically disordered proteins have been attempted in other published studies to model abnormal protein phase transitions related to neurodegenerative diseases. Therefore, the methodology behind the present study lacks novelty in the field of neurodegenerative diseases in general. Authors should site the original work that developed optoDroplet method (PMID: 28041848). In this regard, the last sentence of Abstract "Collectively, our findings provide a unique tool for the generation, visualization, and dissection of the role of protein aggregation in neurodegeneration" is overstated and need to be more specific. Also, the title needs to be more specific, too, as the basic principle has been published.

2: CRY2-based optogenetic method can temporally control protein-protein interaction, but it is still difficult to identify which intermediate species that form during the assembly of oligomer and aggregates are cytotoxic. As mentioned in the discussion, it is still possible that the LIPA-α-syn aggregates are the product of neuroprotective response of cells. The authors need to weaken the claim like, "In the present study, we reported that α-syn aggregation induced a significant loss of DA neurons in the midbrain that was correlated with motor dysfunction." (p17). Also, regarding the first sentence of Abstract "Neurodegenerative disorders refer to a group of diseases triggered by the aggregation of normal proteins into proteinaceous inclusions.", it is not known whether neurodegeneration is "triggered" by protein aggregation.

3: It seems that LIPA-α-syn aggregation (LLPS?) first takes place in the nucleus immediately after light illumination and cytoplasmic aggregation comes after (Sup Video2). The nuclear aggregation/LLPS is also detectable in LIPA-α-syn∆NAC, but cytoplasmic aggregation did not follow (Sup Video3). The nuclear aggregation appears less evident in LIPA-Empty (Sup Video1). Is it possible that LIPA-α-syn phase transition occurs in a step-wise manner in different subcellular compartments? It would be interesting to know what the subnuclear structure of LIPA-α-syn that appears right after the illumination is. The authors at least mention to this LIPA-α-syn-positive subnuclear structure, even if they think these are not important.

Reviewer #2, Srini Subramaniam: This study developed a new tool that is light-induced protein aggregation (LIPA) of a-synuclein protein and characterized its Parkinson disease-like phenotype in a rodent model.

Although it is established that a-synuclein aggregation is a contributing factor in PD, the mechanisms of protein aggregation are unclear. Moreover, despite its ubiquitous presence, how a-synuclein exerts its effect on mid-brain neuron degeneration is unknown. The article by Berard develops optogenetic-combined tools to elicit a-synuclein aggregation to study the spatial and temporal aspects of a-synuclein aggregation and, more importantly, its role in PD-like development.

The article contains state-of-the-art techniques and largely well-controlled experiments with high rigor. I recommend its publication, and however, I have the following recommendations and concerns.

Results: 

Although all the necessary controls are included, I wonder why the light control is not included. Can a-Syn-(WT) mCherry alone induce aggregate upon blue light exposure? 

See the related question below.

In Figure 1F, please described the loading control how it is determined (I guess nitrocellulose retained all of the protein).

Figure 2C-H. LIPA-a-Syn inclusions using TEM are not clear. Where are the aggregates? The legends (arrow etc.,) appear to show the organelles. The LIPA-a-Syn aggregates presence Figure 2C-H are unclear. The author may have to include TEM images of LIPA-a-Syn mutant control to distinguish the aggregates.

Figure S4, STEM images should also include LIPA-empty or LIPA-a-Syn to appreciate the light-induced aggregate distribution.

Figure 3. It is unclear whether a-Syn-GFP forms aggregates due to light and how does a-Syn-GFP alone (without LIPA-a-Syn) look when exposed to blue light. Alternatively, LIPA-a-Syn mutant control in the seeding experiments in Figure 3D would be beneficial.

It is related to the above question, what percentage of LIPA-a-Syn turned into aggregates upon blue light exposure? Data show (Fig 2 B, D) that the blue light effect is time-dependent, but it is not clear that blue light ultimately converts all the soluble a-Syn into aggregated form.

Discussion.

First paragraph "Survival of striatal neurons." I guess this should be mid-brain DA neurons.

"LIPA-induced α-syn aggregates precipitate dopaminergic neuronal loss and induce parkinsonian- like symptoms."

Under this section, authors discuss Lewy body (LB)'s role as protective vs. toxic and proclaim their data indicates LB-like LIPA-a-Syn aggregates are toxic. LIPA-a-Syn in their system may exist as LB as well as oligomeric species, which are considered harmful. Authors own data, western blot Figure 3 indicates besides higher-order (non-gel penetrating, top of the gel) LIPA-a-Syn species, there is also oligomeric (gel-entered, middle of the gel) LIPA-a-Syn species.

So, soluble and oligomeric species of LIPA-a-Syn might be causing toxicity, not the LB-like aggregate species. Contemplating this possibility in the discussion may be helpful.

Reviewer #3: Bérard, Sheta et al provide an interesting and thought-provoking report on a novel potential methodology of modeling aSyn aggregation and toxicity in cells and in vivo.

The method, in brief, is based on light-induced aggregation of aSyn. The authors demonstrate inclusion formation in the absence of toxicity in cultured cancerous cells.

In vivo, the authors report toxicity in the affected brain regions upon AAV delivery and light induction of aggregation. The concept of pathogenic spread is also touched upon. 

Similar work had been done for TDP43 and APP, for aSyn it is novel. If used successfully by this group and others to answer biological questions, the method could turn out to be of significance.

Statistical analyses seem okay, supplementary information and method details are acceptable.

For the study to be considered for publication, the authors should address the following major points:

Fig. 1: Without details on the expression levels of all transgenes, this figure is inconclusive. Fig. 1F may serve that purpose, but it seems like a rather unusual assay. The linearity of the assay is not obvious and a negative control is missing. SDS-PAGE and anti-mCherry blot would be more convincing because the kDa resolution would also allow the reader to assess degradation problems, SDS-stable aggregates etc. Fig. 3F, 3rd lane, suggests aSyn-LIPA may express higher than empty-LIPA. If that is the case, the experiment is inconclusive.

Fig. 2: This figure would be a lot stronger with more controls. Cells containing LIPA-empty inclusions as shown in Fig. 1 would be very good controls. The more transient, presumably not beta-sheet-rich, LIPA-empty inclusions should stain negative for ThioS and not overlap with the other markers as well. Using this control in a consequent manner would also dissipate any doubt about the true nature of the signals (bleed-through from other channels etc.) The authors are strongly encouraged to add this control. It would also be good to see what LIPA-empty inclusions look like by EM.

There also seems to be some disconnect between the apparent lack of fibrils by EM and the ThioS staining. Wouldn't one expect that sth that stains clearly for ThioS is largely fibrillar in the cell? If not: are smaller aggregates sufficient for ThioS staining? What is the nature of the aggregates is definitely a key question here... The SDS-stable aggregates in Fig. 3E are low in abundance. The fibrillar nature of the purified material is not beyond any doubt since it could be a post-lysis artefact. Shahmoradian et al actually suggested that LBs are largely devoid of fibrillar material, and your data are in line with that, which is actually quite striking. The ThioS data, however, may be in contrast to this. In my own experience, and talking to colleagues over the years, using ThioS staining on cells is tricky and needs very good controls to be convincing... ThioS may also bind to vesicle aggregates due to its hydrophobic nature.

Fig. 3: It would be great to see the results of the following experiment: Cells expressing LIPA-empty or LIPA-aSyn plus minus PFF, plus minus light. PFFs should seed LIPA-aSyn just fine even without light. LIPA-empty should be unaffected. It'd be very interesting to see the effect of light. Fig. 3D is very impressive. It is also striking that in Fig. 3 aggregated aS was indeed taken up. 

Fig. 4/5: These are quite impressive, complicated experiments. Yet, controls would make the data more conclusive. What is shown - please correct me if I am wrong - is consistent with the simple notion that expressing a stressor in a certain brain area causes problems in that brain area. Expressing a control fusion (TDP43-LIPA?) in the same regions leading to different results would help establish aSyn-specificity and PD relevance better. It would also be good to know if the transgenes are sorted correctly to synapses, and what happens to the endogenous protein when the transgene is there and caused to aggregate. Mouse aSyn slows down human aS aggregation, but would mouse aSyn still be seeded to aggregate? Would human-aSyn-LIPA induced aggregates spread to other neurons to cause mouse aSyn to aggregate? Does LOF of endogenous mouse aSyn play any role here, or is it all GOF of human LIPA-aSyn?

General comments:

The presence of the large mCherry tag could be addressed better. This may affect aSyn fibrillization etc. The method doesn't really need the tag, unless for visualization. Have you considered to do a subset of expts without the tag?

Not least due to the tag, plus the forced dimers as subunits, statements about "authentic LB formation" may have to be toned down. The nature of LBs is under debate, and it is not clear if the inclusions here are closely similar to aSyn in LBs.

Carrying deltaNAC and empty controls more through the paper, plus adding a control of another aggregation-prone protein to establish aSyn/PD specificity could have added more clarity.

Minor: Please correct "meduim" in Fig. 2I. Please check for typos throughout.

Reviewer #4: In this manuscript, Berard et al. describe a novel system for inducing the aggregation of alpha-synuclein (A-syn), a major protein constituent of Lewy bodies in synucleinopathies. They apply here a light-induced protein aggregation (LIPA) paradigm whereby a Cry2/A-syn/mCherry construct is expressed and aggregation induced using blue light. The data provided nicely indicate the formation of insoluble intracellular bodies in transfected cell lines and, importantly, human iPSC neurons. Such inclusions show many of the biochemical features associated with human brain Lewy bodies such as serine 129 phosphorylation, thioflavin positivity, ubiquitylation, and colocalization with Hsp70. TEM confirmed these to be vesicle-rich as recently described for Lewy bodies, although classical A-syn fibrils were conspicuously absent. Despite that, A-syn extracted from LIPA+light cells were able to induce further aggregation in multiple assays (RTQuic, PMCA and in other cells), suggestive of fibrillar species within, albeit at low quantities. The authors then demonstrate in vivo application of these LIPA constructs in the substantia nigra and show that blue light treatment induced inclusion formation that was associated with TH-neuron loss and reduced/impaired activity in the striatum and manifested as motor deficits. Overall, this is an interesting study that introduces a novel tool to the arsenal for investigating A-syn aggregation. The experiments are logical and generally well laid out. On the other hand, the data presented raise some questions about the nature of the intracellular bodies that are formed using this approach. Although the authors rightly point out that they have many biochemical features of Lewy bodies, they also differ in a number of ways, most noticeably in terms of their morphology and lack of prominent fibril-like structures. The ability of the LIPA induced A-syn to further replicate via cell to cell transfer was also not tested in culture or in vivo. Clarifying these issues and addressing the more moderate points below would enhance this manuscript.

Major points:

- Although the authors provide compelling data that LIPA A-syn inclusions disrupt share some biochemical traits with Lewy bodies, the morphological contrast between the two is not discussed in much detail. With the exception of some of the HEK cells in Fig 1,the LIPA inclusions tend to be small and numerous (and lacking obvious fibrils) whereas Lewy bodies are typically large with 1-2 present per neuron. The resemblance with Lewy bodies should therefore be toned down in this context.

- The in vivo data in Figs. 4 and 5 are highly interesting but omits some quantitative details. For example, what was the distrubution of LIPA A-syn aggregates and what % of TH neurons developed them? Was there spread detected in areas outside of the midbrain and were aggregates induced in any non-neuronal cells? Only individual cells are shown in most panels so it is hard to discern this. 

Minor points:

- In several places in the manuscript, the authors refer to "monomeric A-syn-GFP" (e.g. page 10, 2nd paragraph) when describing recruited A-syn, which may be (but more likely not) in the monomeric form. This should be modified to simply "A-syn-GFP" to avoid confusion.

- Image in Fig 5B is very hard to make out and was not sufficient for this reviewer to evaluate. Please include one with higher contrast and resolution.

- Figures 5G-I: the control graphs from Fig S8 should be included here in the main figure as well. According to the color scale provided, LIPA+light appears to increase the amplitude of activity within the first 100 sec (Fig 5F middle panel). If that is the case, this should be mentioned in the text.

---

## [Editor Report · Decision Letter 2]

9 Feb 2022

Dear Dr Oueslati,

Thank you for submitting your revised Methods and Resources article entitled "A light-inducible protein clustering system for modeling α-synuclein aggregation in Parkinson’s disease" for publication in PLOS Biology. I have now discussed your new version with the Academic Editor and am pleased to let you know that we will probably accept this manuscript for publication, provided you satisfactorily address the following data and other policy-related requests:

1) Title: We would like to suggest a minor modification: "A light-inducible protein clustering system for in vivo analysis of α-synuclein aggregation in Parkinson’s disease."

2) Blurb: Please provide a blurb which (if accepted) will be included in our weekly and monthly Electronic Table of Contents, sent out to readers of PLOS Biology, and may be used to promote your article in social media. The blurb should be about 30-40 words long and is subject to editorial changes. It should, without exaggeration, entice people to read your manuscript. It should not be redundant with the title and should not contain acronyms or abbreviations. For examples, view our author guidelines: https://journals.plos.org/plosbiology/s/revising-your-manuscript#loc-blurb

3) Ethics: Please include in your manuscript the ID number of your protocols approved by the Animal Welfare Committee of Université Laval.

4) Data: You may be aware of the PLOS Data Policy, which requires that all data be made available without restriction: http://journals.plos.org/plosbiology/s/data-availability. For more information, please also see this editorial: http://dx.doi.org/10.1371/journal.pbio.1001797

Note that we do not require all raw data. Rather, we ask for all individual quantitative observations that underlie the data summarized in the figures and results of your paper. For an example see here: http://www.plosbiology.org/article/info%3Adoi%2F10.1371%2Fjournal.pbio.1001908#s5

These data can be made available in one of the following forms:

I) Supplementary files (e.g., excel). Please ensure that all data files are uploaded as 'Supporting Information' and are invariably referred to (in the manuscript, figure legends, and the Description field when uploading your files) using the following format verbatim: S1 Data, S2 Data, etc. Multiple panels of a single or even several figures can be included as multiple sheets in one excel file that is saved using exactly the following convention: S1_Data.xlsx (using an underscore).

II) Deposition in a publicly available repository. Please also provide the accession code or a reviewer link so that we may view your data before publication.

Regardless of the method selected, please ensure that you provide the individual numerical values that underlie the summary data displayed in the following figure panels: Figures 1CDEFHI, 3ABCHIJ, 4F-K, 5F-H, S1AB, S2B, S7BCDEK, S8BDFG, and S10.

4.1) Please also ensure that each figure legend in your manuscript includes information on where the underlying data can be found and that your supplemental data file/s has/have a legend.

4.2) Please ensure that your Data Statement in the submission system accurately describes where your data can be found.

We expect to receive your revised manuscript within two weeks. 

*Published Peer Review History*

*Early Version*

Sincerely,

Gabriel

Gabriel Gasque, Ph.D.,

Senior Editor,

ggasque@plos.org,

PLOS Biology

---

## [Editor Report · Decision Letter 3]

18 Feb 2022

Dear Dr Oueslati,

On behalf of my colleagues and the Academic Editor, Gillian P. Bates, I am pleased to say that we can in principle accept your Methods and Resources article "A light-inducible protein clustering system for in vivo analysis of α-synuclein aggregation in Parkinson’s disease" for publication in PLOS Biology, provided you address any remaining formatting and reporting issues. These will be detailed in an email that will follow this letter and that you will usually receive within 2-3 business days, during which time no action is required from you. Please note that we will not be able to formally accept your manuscript and schedule it for publication until you have any requested changes.

***IMPORTANT: As you address these requests, please ensure that the supplemental data file has a legend.

PRESS

Sincerely, 

Gabriel Gasque, Ph.D. 

Senior Editor 

PLOS Biology

ggasque@plos.org